# Cooperation of Striatin 3 and MAP4K4 promotes growth and tissue invasion

Jessica Migliavacca[1], Buket Züllig[1], Charles Capdeville[1], Michael A. Grotzer[2] & Martin Baumgartner [1✉]

MAP4K4 is associated with increased motility and reduced proliferation in tumor cells, but the regulation of this dichotomous functionality remained elusive. We find that MAP4K4 interacts with striatin 3 and 4 (STRN3/4) and that STRN3 and MAP4K4 exert opposing functions in Hippo signaling and clonal growth. However, depletion of either STRN3 or MAP4K4 in medulloblastoma cells reduces invasion, and loss of both proteins abrogates tumor cell growth in the cerebellar tissue. Mechanistically, STRN3 couples MAP4K4 to the protein phosphatase 2A, which inactivates growth repressing activities of MAP4K4. In parallel, STRN3 enables growth factor-induced PKCθ activation and direct phosphorylation of $VASP_{S157}$ by MAP4K4, which both are necessary for efficient cell invasion. $VASP_{S157}$ directed activity of MAP4K4 and STRN3 requires the CNH domain of MAP4K4, which mediates its interaction with striatins. Thus, STRN3 is a master regulator of MAP4K4 function, and disruption of its cooperation with MAP4K4 reactivates Hippo signaling and represses tissue invasion in medulloblastoma.

[1] Pediatric Molecular Neuro-Oncology Research, Division of Oncology, Children's Research Center, University Children's Hospital Zürich, Zürich, Switzerland.
[2] Division of Oncology, University Children's Hospital Zürich, Zürich, Switzerland. ✉email: Martin.Baumgartner@kispi.uzh.ch

Medulloblastoma (MB) is the most common malignant brain tumor in children[1]. Four distinct molecular subgroups (Wingless (WNT), Sonic Hedgehog (SHH), group 3, and group 4) can be discriminated with a total of 12 subtypes[2–4]. The characteristic tendency of MB cells to locally infiltrate the cerebrospinal fluid (CSF) via leptomeningeal dissemination reduces the efficacy of current treatments such as surgery and local radiotherapy[5]. One key driver of tissue invasion and growth in MB is the Fibroblast Growth Factor Receptor (FGFR) signaling pathway[6,7]. These FGFR functions rely on complex orchestration of parallel, potentially opposing signaling pathways to overcome the incompatible concomitant induction of cell migration and proliferation. The molecular mechanisms that balance FGFR-driven signaling towards tissue invasion and growth in MB and other tumors remain incompletely understood, and no specific therapy is available to subvert this balance towards tumor suppression[8]. Improved anti-dissemination therapies should specifically repress both the pro-migratory and the proliferation-promoting signaling in tumor cells. The aim of this study was to address the concomitant regulation of proliferation and invasiveness downstream of the activated FGFR, to delineate the underlying molecular mechanisms, and to identify novel potential drug targets in MB.

The serine/threonine mitogen-activated protein kinase kinase kinase kinase 4 (MAP4K4/HGK/NIK), is upregulated in MB patient samples and acts as a critical promoter of receptor tyrosine kinase (RTK)-induced MB cell dissemination[9,10]. MAP4K4 controls a wide range of biological processes[11–16], which can be explained by its interaction with a number of structurally and functionally different effectors. The pro-migratory function of MAP4K4 is in part due to its capability to directly phosphorylate the ezrin, radixin, moesin (ERM) proteins, which mediate lamellipodium formation in response to growth-factor (GF) stimulation[17]. MAP4K4 couples GF signaling to actin polymerization through phosphorylation of the actin-related protein 2 (Arp2) subunit of the Arp2/3 complex, which increases its actin nucleation activity and triggers membrane protrusion in response to EGF[18]. In MB, MAP4K4 promotes turnover and activation of the receptor tyrosine kinase c-MET and of the β-1 integrin adhesion receptor, which is necessary for membrane protrusion at the leading edge of invading cells[10]. The pleiotropic functionality of MAP4K4 and its high expression in the healthy brain tissue argue against global pharmacological repression of its catalytic activity. Rather, more subtle interference strategies that target only subsets of its functions would be needed to repress MAP4K4 oncogenic activities specifically.

The related kinase Misshapen-like kinase (MINK)[19] and MAP4K4[20,21] were identified as components of the Striatin-interacting phosphatase and kinase (STRIPAK) complex[22–24]. The STRIPAK complex is a supramolecular scaffold assembled around the striatin family proteins (STRN, STRN3, STRN4). STRNs act as regulatory subunits of the protein phosphatase PP2A and are associated with STRN-interacting proteins (STRIP1/2) and with members of the germinal center kinase (GCK) families[22,25]. The STRIPAK complex controls Hippo tumor suppressor signaling towards YAP/TAZ transcriptional activation and proliferation by repressing associated Hippo-activating kinases including MAP4K4. In this context, STRIPAK-MAP4K4 interaction contributes to small T antigen-mediated cell transformation[21]. MAP4K4 interacts with STRN4, which orchestrates MAP4K4 repression and thereby causes decreased Hippo signaling[20]. MAP4K4 interaction with STRN4 was also proposed to promote malignant characteristics of cancer cells[26]. However, the mechanistic details how STRIPAK-controlled MAP4K4 contributes to tumorigenesis, in particular migration and tissue invasion, remain elusive.

We used proximity-dependent biotin identification (BioID) and kinase activity profiling to functionally explore the MAP4K4 interactome in tumor cells. We specifically addressed how MAP4K4 and STRIPAK complex components integrate FGFR-driven oncogenic signaling towards tumor growth, dissemination, and tissue invasion. Our study identified the MAP4K4-STRIPAK complex as a central hub and bifurcation point, which promotes tissue invasion and growth in MB tumors. It provides relevant mechanistic insights in the regulation of these central processes in MB tumor cells, and it runs against the recently established dogma of STRIPAK-MAP4K4 antagonism by demonstrating a cooperative activity of STRN3 and MAP4K4 towards increased invasiveness via the activation of PKCθ and VASP.

## Results

**The CNH domain of MAP4K4 mediates STRIPAK complex interaction in MB tumor cells.** We used BioID to assess the MAP4K4 interactome and identified proteins biotinylated by N- or C-terminally biotin ligase-fused (BioID2[27]) full-length MAP4K4 (FLAG-BioID2-MAP4K4 and MAP4K4-BioID2-FLAG) or BioID2-FLAG alone through mass spectrometry (MS)[28,29] in HEK-293T cells (Fig. S1A). The expression of 3xFLAG-tagged MAP4K4-BioID2 was confirmed by anti-FLAG immunoblot (IB) analysis and of biotinylation activity by horseradish peroxidase (HRP)-coupled streptavidin binding (Figure S1B). Biotinylated proteins were enriched by affinity-capturing using streptavidin-conjugated beads and subjected to MS. We identified a total of 156 MAP4K4-specific proteins (Fig. S1C and Supplementary Data 1). Gene ontology (GO) analysis with these proteins predicted functions such as endosomal transport[10], actin polymerization[18], or regulation of cell adhesion[16] (Fig. 1a), and found enrichment of components of the STRIPAK complex (Fig. 1a–c). We confirmed the interaction of 3xFLAG-BioID2-MAP4K4 with STRN3, STRN4, and STRIP1 in HEK-293T cells by co-immunoprecipitation analysis (Fig. 1d). In agreement with other studies[21], these data confirm MAP4K4 association with the STRIPAK complex in cells, possibly through direct interaction with STRN4 and STRN3.

To determine the MAP4K4 interactome in MB tumor cells, we generated cell lines stably expressing either the BioID2-MAP4K4 fusion proteins or BioID2 alone (Fig. S1D). BioID2-MAP4K4 localized to the cytoplasm, the primary residency of MAP4K4, where it co-localized with biotinylated proteins (Fig. S1E). In contrast, BioID2 distributes throughout the cells, with enrichment in the nucleus. Streptavidin affinity purification-MS and anti-FLAG immunoprecipitation confirmed the association of MAP4K4 with the STRIPAK complex in MB cells (Fig. 1e, f, Fig. S1F, and Supplementary Data 2). MAP4K4 is dispensable for STRNs interaction with STRIP1, while STRN3 is necessary for MAP4K4 association with the complex (Fig. 1g). Unlike other germinal center kinases that have been reported to interact with STRIP1 in an activity-dependent manner[30], inhibition of MAP4K4 by GNE-495 did not impact its constitutive association with STRN4, STRN3, and STRIP1 (Fig. S1G). We generated DAOY cells stably expressing FLAG-tagged single domains or truncated versions of MAP4K4 (Fig. 1h). We found that N-terminally truncated MAP4K4 (ID1-ID2-CNH, 290-1273) pulled down STRN4 and STRIP1 more effectively than C-terminally truncated KD-ID1-ID2 (1-954) (Fig. 1h, i). The kinase domain (KD, 1-289) or the intermediate domains (ID1, 290-634 or ID2, 635-954) alone are not sufficient for pulling down STRN4 or STRIP1. Combined, these results suggest that the citron homology domain (CNH, 954-1273) is required for MAP4K4-STRIPAK interaction in MB tumor cells.

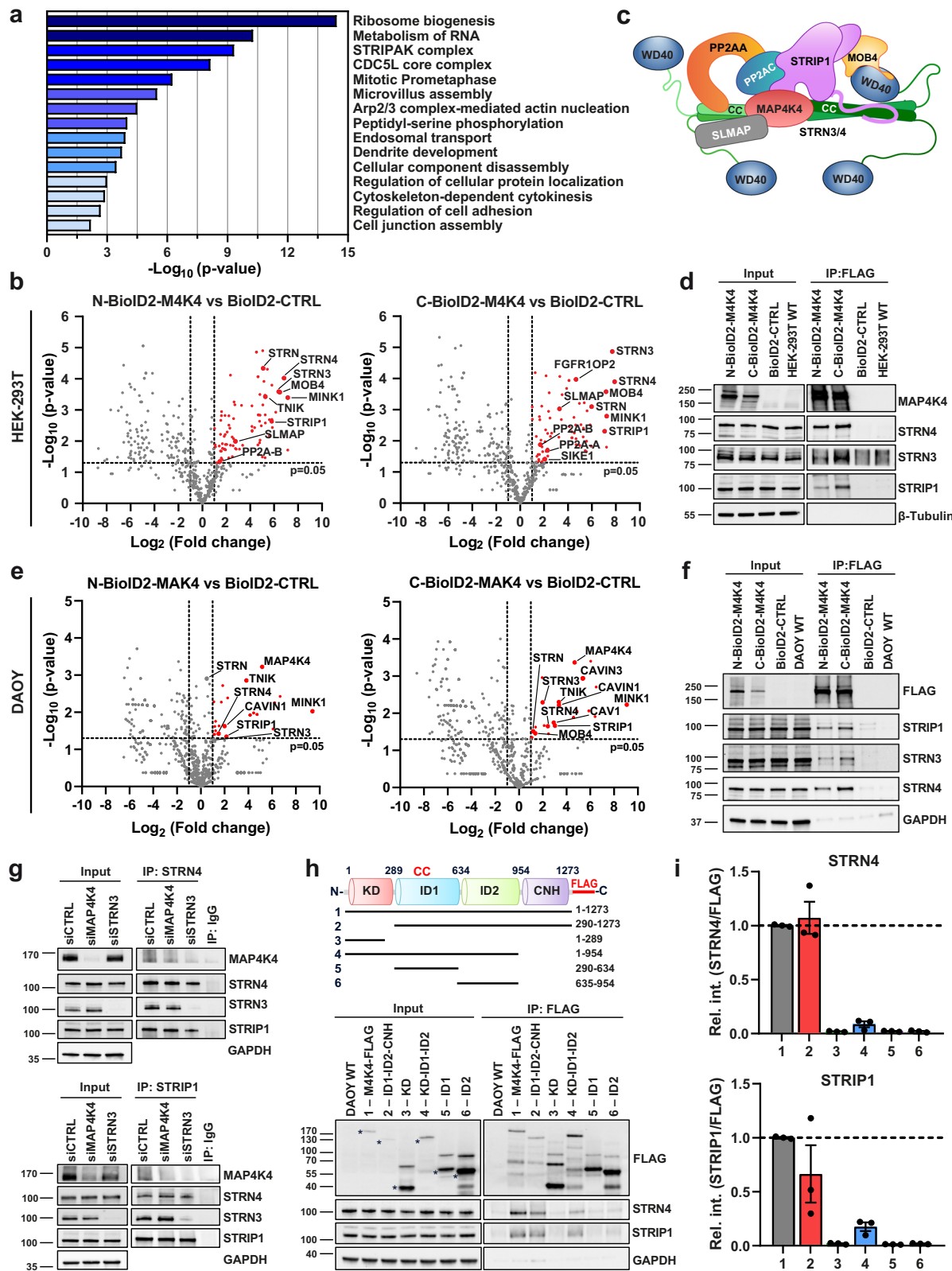

**STRN3 and STRN4 contribute to bFGF-induced collagen I invasion of MB cells**. STRIPAK complex members STRN4, STRN3, and STRIP1 are highly expressed in MB, and the expression of STRNs correlates positively with site-specific phosphorylation of MAP4K4 in ID1 and negatively with phosphorylation in ID2 in vivo (ref. [31] and Fig. S2A–D). This suggests that the interaction of these proteins contributes to MAP4K4

regulation and MB pathogenesis, possibly by controlling differential phosphorylation of MAP4K4 (Fig. S2E). MAP4K4 promotes GF-induced MB cell motility[9,10]. Hence, we tested the implication of STRIPAK complex proteins in invasive motility in the SHH MB line DAOY by stable knockout (KO) of STRN4, STRN3, or STRIP1 (Fig. S3A, B). Using the spheroid collagen I invasion assay (SIA[32]), we observed that STRN4 KO abrogated

**Fig. 1 BioID identifies MAP4K4-STRIPAK complex interaction in MB cells. a** Gene ontology (GO) classifications of MAP4K4-interacting protein from BioID experiments conducted in HEK-293T. Mass spectrometry was performed after immunoprecipitation (IP) with streptavidin-conjugated beads of N- or C- BioID2-MAP4K4 and BioID2 alone overexpressing cells. **b** Volcano-plots of interacting proteins in N-BioID2-MAP4K4 (left) or C-BioID2-MAP4K4 (right) HEK-293T cells compared to BioID2-CTRL. $Log_2$ ratios of cumulated peptide detection values plotted against negative $Log_{10}$ p-values of Student's t-test. Enriched proteins are highlighted in red ($p < 0.05$ and > 2-fold enrichment) and members of the STRIPAK complex indicated. $n = 3$. **c** Architecture of the STRIPAK complex, adapted from[72]. **d** Immunoblot (IB) detection of BioID2-MAP4K4-FLAG and components of STRIPAK complex in anti-FLAG-IP HEK-293T cells. **e** Volcano-plots of protein interactions identified by streptavidin affinity purification-MS in N-BioID2-MAP4K4 (left) or C-BioID2-MAP4K4 (right) DAOY cells compared to BioID2-CTRL. Members of the STRIPAK complex are highlighted. $n = 2$. **f** IB of MAP4K4-FLAG and components of STRIPAK complex after anti-FLAG-IP in DAOY cells. **g** IB analysis of MAP4K4 and components of STRIPAK complex after anti-STRN4 (top) or anti-STRIP1 (bottom) immunoprecipitation in DAOY cells transfected for 48 h with the indicated siRNA. Normal rabbit IgG was used as control for the IP. **h** Top: MAP4K4 domain architecture and MAP4K4 constructs used. Numbers represent the amino acids corresponding to full-length MAP4K4. KD, kinase domain; ID, intermediate domain; CNH, citron homology domain; CC, coiled-coil. Lower: Co-IP and IB analysis of STRN4 and STRIP1 in DAOY cells transduced with lentiviral vectors expressing full-length or truncated MAP4K4-FLAG. Asterisks indicate relevant bands. **i** Quantification of co-IP from H ($n = 3$, means ± SEM). See also Fig. S1. Source data for quantifications shown in (**a**), (**b**), (**e**), and (**i**) are available in Supplementary Data 5.

bFGF-induced cell invasion in DAOY cells, while KO of either STRIP1 or STRN3 had little or no effect (Fig. S3C).

We next determined the effect of simultaneous depletion of various STRIPAK components on GF-induced migration using siRNA (Fig. S3D, E) in DAOY cells. Depletion of STRN4 confirmed its implication in bFGF-induced DAOY collagen I invasion, and depletion of STRN3 by siRNA also reduced bFGF-induced invasion (Fig. 2a). In contrast, depletion of STRIP1 caused a marked increase in both EGF- and bFGF-induced dissemination. Consistent with a functional redundancy of STRN3 and STRN4, we observed that combined depletion of STRN3 and STRN4 had an additive effect on EGF- or bFGF-induced collagen I invasion. Thus, STRN3 and STRN4 are, together with MAP4K4, required for GF-induced collagen I invasion in MB. We tested the implication of STRIPAK components in migration control with another SHH MB cell line, UW228, an confirmed that depletion of either MAP4K4, STRN4, or STRN3 reduced bFGF-induced collagen I invasion (Fig. 2b and Fig. S3F, G).

STRIPAK complex can regulate associated kinases by enabling dephosphorylation through the PP2A catalytic subunit[33,34], and inhibition of PP2A sensitizes tumors to radiation and/or chemotherapy and attenuates cell migration[35,36]. We found that the PP2A inhibitor LB-100 decreased bFGF-induced collagen I invasion in a dose-dependent manner (Fig. 2c) without affecting cell viability (Fig. S3H). Thus, STRN3/4 enables or facilitates the pro-invasive phenotype of GF-activated MB cells, and PP2A does not negatively regulate invasiveness under these conditions.

**STRN3 and MAP4K4 cooperate towards tumor cell expansion in the tissue context.** We used an ex vivo organotypic cerebellum slice culture model (OCSCs) to test STRIPAK implication in MB tissue invasion (Fig. S4A). OCSCs maintain key features of the normal cerebellum cytoarchitecture and constitute a physiologically relevant microenvironment model[37], where FGFR signaling contributes to MB growth and tissue invasion[6]. We first tested whether the depletion of MAP4K4 or STRIPAK components prevented growth and tissue invasion of DAOY cells. We found that STRN3 depletion caused a significant reduction of tumor area (Fig. 3a). Treatment of the co-culture with EGF enhanced brain tissue invasion of the tumor cells (Fig. S4B, C). Depletion of MAP4K4 caused rounded cell morphology and moderately reduced EGF-induced invasion (Fig. S4B–D). Depletion of STRN3 caused compaction of the tumor cell spheroids and prevented tissue invasion also under EGF-stimulation (Fig. S4C, D). Conversely, depletion of STRIP1 induced a more infiltrative behavior, similar to what was observed in vitro (Fig. 2a). The combined depletion of MAP4K4 and STRN3 in DAOY cells caused a dramatic reduction of tumor area, both under basal condition (Fig. 3a, b) and under EGF stimulation (Fig. S4B–D). The reduced tumor area in cells with combined depletion is not the result of impaired proliferation, as remaining cells display proportionally similar EdU incorporation as the control (Fig. S4E).

To confirm STRN3 implication in tissue invasion, we implanted HD-MBO3 group 3 MB cell lines KO for either MAP4K4, STRN4, STRN3, or STRIP1 (Fig. S3B) in OCSCs. HD-MBO3 cells are invasive and display a high growth rate[37]. We observed a moderate for MAP4K4 KO and a significant reduction in the expansion for STRN3 KO cell-derived tumors, both after five days in basal or GF-stimulated conditions (Fig. 3c, d and Fig. S5). Moreover, the number of EdU-positive nuclei was considerably reduced in sgSTRN3 and sgSTRN4 but not in sgMAP4K4 cells (Fig. 3e).

This data highlights the role of the STRN3 protein in growth and invasion in the tissue context and reveal a cooperative functionality of MAP4K4 and STRN3 towards tumor expansion.

**Opposing functions of STRN3/4 and MAP4K4 in cell proliferation and YAP/TAZ target gene regulation.** Depletion of STRN3 but not of MAP4K4 reduced tumor cell growth in the cerebellum tissue. We, therefore, tested the implication of MAP4K4 and of STRIPAK complex components in colony-formation and clonal growth. Depletion of MAP4K4 caused a marked increase in colony number independent of growth factor stimulation in DAOY cells (Fig. 4a). Conversely, depletion of STRN3 impaired colony formation and clonal growth in DAOY cells, and also caused a significant decrease in the number of colonies in HD-MB03 cells (Fig. 4b), demonstrating that STRN3 was necessary for clonal growth and colony formation in MB cells. The effect of STRN3 depletion was particularly striking under bFGF stimulation in HD-MB03 cells, which promoted massive colony-forming activity in this group 3 MB model (Fig. 4b).

MAP4K4 activates the Hippo tumor suppressor pathway and thereby represses YAP/TAZ target gene expression and growth[38,39]. By antagonizing this function of MAP4K4, the STRIPAK complex promotes YAP/TAZ transcriptional activity[21,40,41]. We found that MAP4K4 depletion significantly increased the expression of the YAP/TAZ target genes *CTGF*, *CYR61*, and *ANKRD1*, which supports a Hippo-pathway activating function of MAP4K4 in MB cells (Fig. 4c, d). Conversely, depletion of STRN3 or STRN4 decreased transcription of these genes (Fig. 4c, d). Thus, the increased clonal growth in MAP4K4-depleted MB cells could be a consequence of the failure of these cells to activate Hippo signaling, and the decreased clonal growth in STRN3-depleted cells a consequence of de-repressed Hippo signaling (Fig. S6A). Interestingly, the repression of YAP/TAZ

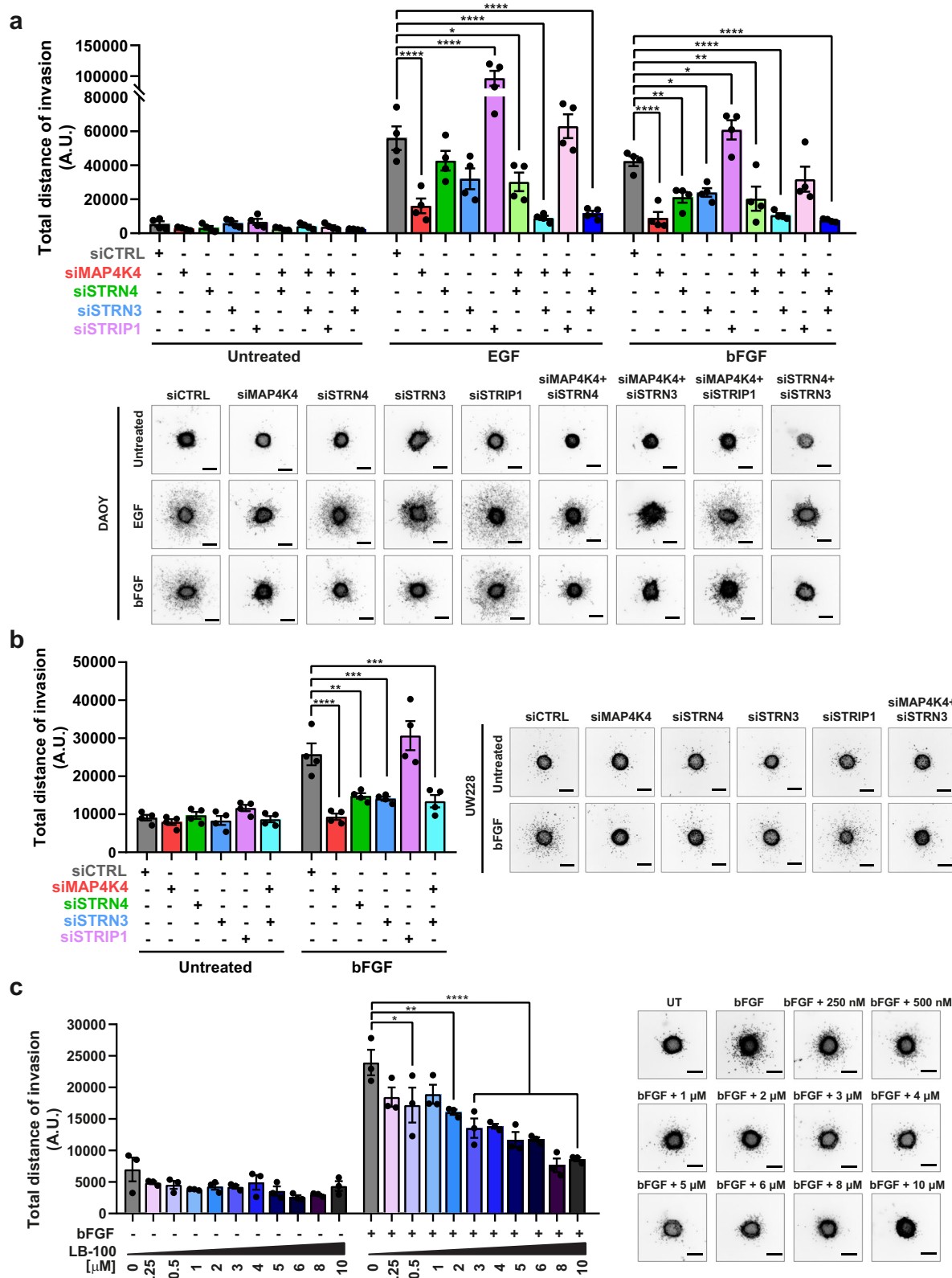

**Fig. 2 STRIPAK complex disruption impairs GF-induced collagen I invasion. a** Invasion and representative images of 3D spheroid invasion assay (SIA) of DAOY cells transfected with 10 nM of the indicated siRNA ± GFs (EGF: 30 ng/ml, bFGF: 100 ng/ml). Invasion is calculated as sum of the distances of each migrating cell from the center of the spheroids ($n = 4$, means ± SEM). Scale bar: 300 μm. **b** SIA of UW228 cells transfected with 10 nM of the indicated siRNA ± 100 ng/ml bFGF ($n = 4$, means ± SEM). Scale bar: 300 μm. **c** SIA and representative images of DAOY cells ± 100 ng/ml bFGF and ± increasing concentrations of PP2A inhibitor LB-100 ($n = 3$, means ± SEM). *$p < 0.05$, **$p < 0.01$, ***$p < 0.001$, ****$p < 0.0001$ (one-way ANOVA). Scale bar: 300 μm. See also Fig. S3. Source data for quantifications shown in (a-c) are available in Supplementary Data 6.

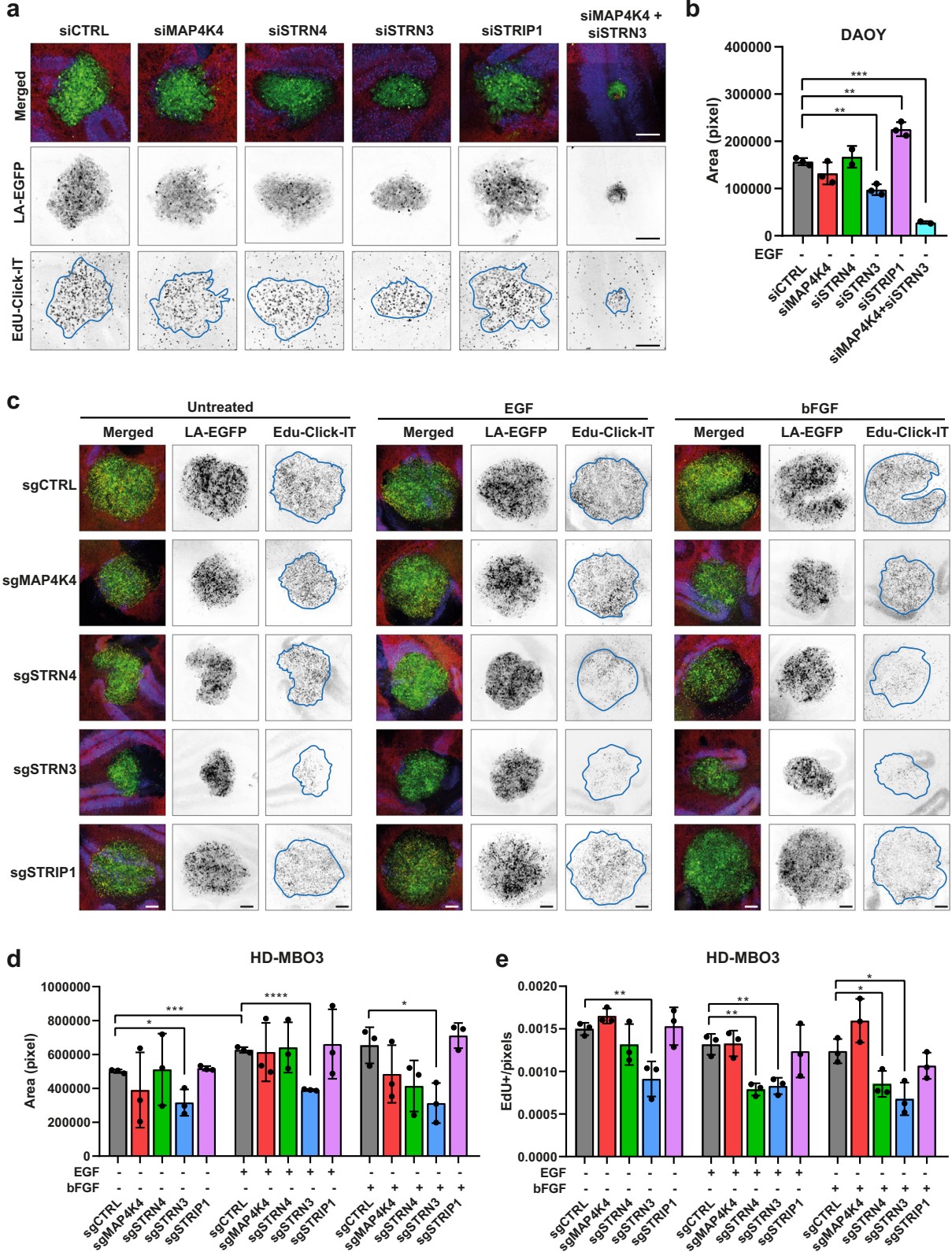

target genes induced by STRN3 or STRN4 depletion was rescued to control levels by the concomitant depletion of MAP4K4 (Fig. 4c).

The STRIPAK phosphatase PP2A antagonizes Hippo kinases and promotes a YAP/TAZ transcriptional program[21,42]. Consistently, PP2A inhibition by LB-100 repressed transcription of YAP/TAZ target genes and exactly phenocopied the depletion of

either STRN3 or STRN4 (Figs. 4e, S6B). Treatment of STRN3- or STRN4-depleted cells with LB-100 did not further decrease YAP/TAZ target gene expression, demonstrating that PP2A exerted its activity exclusively via STRN3 and STRN4. In contrast, depletion of MAP4K4 in LB-100-treated cells restored YAP/TAZ target gene expression to control levels, indicating the implication of a compensatory mechanism controlling YAP/TAZ target gene

**Fig. 3 STRN3 is necessary for invasive growth of SHH and group 3 MB cell model in cerebellar tissue. a** Maximum intensity projections (MIP) of representative confocal sections of organotypic cerebellum slice culture (OCSC) implanted with DAOY tumor cell spheroids for 48 h. Green: Lifeact-EGFP; blue: calbindin (Purkinje cells); red: GFAP. Middle: Inverted greyscale images of MIP of LA-EGFP channel, lower: Inverted greyscale images of MIP of EdU-Click-IT staining. Scale bar: 200 µm. **b** Quantification of LA-EGFP area of DAOY spheroids shown in (**a**) ($n = 3$, means ± SD). **c** OCSC culture implanted with tumor spheroids derived from HD-MBO3 cells KO for MAP4K4, STRN4, STRN3, or STRIP1. MIP of representative slices five days after implantation and ± treatment with 30 ng/ml EGF or 12.5 ng/ml bFGF. Green: Lifeact-EGFP; blue: calbindin (Purkinje cells); red: GFAP; yellow: Edu-Click-IT. Scale bar: 200 µm. **d** Areas of LA-EGFP positive HD-MBO3 spheroids shown in (**c**) ($n = 3$, means ± SD). **e** Number of EdU-positive nuclei normalized by the area in pixels of the tumor spheroids shown in (**c**) ($n = 3$, means ± SD). *$p < 0.05$, **$p < 0.01$, ***$p < 0.001$, ****$p < 0.0001$ (unpaired Student's *t*-test). See also Figs. S4 and S5. Source data for quantifications shown in (**b**), (**d**), and (**e**) are available in Supplementary Data 7.

expression in cells with a complete loss of STRIPAK control of Hippo signaling (Figs. 4e, S6B). We furthermore found that bFGF stimulation increased YAP/TAZ target gene expression. The STRN3-associated PP2A activity is necessary for the bFGF-induced increase in YAP/TAZ target gene expression as latter was completely blocked by LB-100 treatment or STRN3 depletion (Figs. 4f, g, S6C, D). We observed a further increase in YAP/TAZ target gene expression in MAP4K4-depleted DAOY cells stimulated with bFGF (Figs. 4f, S6C), suggesting that MAP4K4 represses YAP/TAZ target gene expression also under bFGF-stimulation, Collectively, these data confirmed on the one hand a growth suppressing role of MAP4K4 in MB tumor cells. On the other hand, the restored YAP/TAZ target gene expression after co-depletion of STRN3 and MAP4K4 also indicated that tumor growth suppression we observed by this treatment in the tissue (Fig. 3a) cannot solely be explained by its impact on YAP/TAZ target gene expression.

**PKCθ is a pro-invasive effector kinase downstream of MAP4K4 and STRN3.** To identify effectors of MAP4K4 and of STRIPAK complex components downstream of FGFR activation, we profiled changes in global kinase activities using the PamGene array technology. FGFR signaling caused significantly increased activities of CDKs, PKC, SRC kinases, and FAK (Fig. 5a and Fig. S7A). Several of these and additional bFGF-induced kinase activities require the expression of MAP4K4 or of STRIPAK complex components (Fig. 5a and Fig. S7A). The most differently regulated Ser/Thr (STK) kinases cluster in the AGC (cAMP-dependent, cGMP-dependent, and PKC) family of kinases (Fig. S7B–E). The activities of several AGC kinases were reduced when MAP4K4, STRN4, or STRN3 were downregulated, and increased in STRIP1-depleted cells. Moreover, STRIP1 depletion increased the activity of a MAP4K4 signature (Fig. 5a), indicating that STRIP1 enables some repression of MAP4K4 in MB cells. Interestingly, depletion of MAP4K4, STRN3, or STRN4, but not of STRIP1, caused a striking decrease in the bFGF-induced activity signature for novel protein kinases C (nPKCs, PKCδ, ε, η, and θ) (Fig. 5a). By IB analysis we could demonstrate that depletion of MAP4K4 or of STRN3 reduced T538 phosphorylation of the catalytic domain of PKCθ[43] (Fig. 5b), which confirms the kinase activity profiling data and suggested that MAP4K4 and STRN3 promote the activation of PKCθ.

Activated nPKCs could constitute druggable effector kinases to repress MB cell dissemination. We tested this possibility using inhibitors specific for PKCθ (PKCθi[44] and Darovasertib[45]). We observed a dose-dependent reduction of bFGF-induced cell dissemination in DAOY cells (Fig. 5c and Fig. S8A). A similar effect was obtained using the PKCδ inhibitor Rottlerin (Fig. S8B)[46]. Notably, the inhibition of classical PKCα, β, γ kinases by Bisindolylmaleimide I (BIM) did not significantly repress MB cell dissemination (Fig. S8C). PKCθ or PKCδ inhibition also resulted in the compaction of DAOY and HD-MBO3 tumor spheroids in the OCSC model and in the reduction

of cerebellum tissue invasion, without affecting MB cell proliferation (Fig. 5d–f and Fig. S8D–F). Depletion of PKCθ by siRNA (Fig. S8G,H) stalled bFGF-induced DAOY collagen I invasion (Fig. 5g) and repressed bFGF-induced cerebellum tissue invasion of DAOY and HD-MBO3 cells (Fig. 5h, i and Fig. S8I, J).

Collectively these data demonstrate that MAP4K4 and STRN3 promote a pro-migratory and invasive phenotype in MB cells through the activation of common downstream effectors, which include novel but not classical PKC family members.

**STRN3 enables direct VASP$_{Ser157}$ phosphorylation by MAP4K4.** Depletion of MAP4K4 in bFGF-stimulated DAOY cells decreased the phosphorylation of multiple STK and PTK consensus phosphorylation sites (Fig. 6a, Fig. S9A, B, and Supplementary Data 3). Many of these alterations were phenocopied by the depletion of STRN3 (Fig. 6a). Conversely, depletion of STRIP1 caused a global increase in the phosphorylation of STK substrates (Fig. 6a and Fig. S9A). Phosphorylation profiles of PTK substrates were similar under all conditions (Fig. 6a and Fig. S9B), with 23 substrates uniquely altered by MAP4K4 or STRN3 depletion (Fig. 6b), including the phospholipase C-γ (PLC-γ)-binding site[47] of FGFR1 (FGFR$_{Y766}$), and the SRC phosphorylation site of AKT (AKT$_{Y326}$)[48] (Fig. 6c). Additionally, we observed the reduction of FAK1$_{Y570/576}$ phosphorylation in siMAP4K4 and siSTRN3, supporting the previously described role of MAP4K4 in mediating FAK protein activation and focal adhesion assembly[10] (Fig. 6c).

Exploring potential direct substrates of MAP4K4, we found that either MAP4K4 or STRN3 depletion decreased the phosphorylation of VASP$_{S157}$ (Fig. 6c), an F-actin elongation promoting factor involved in filopodia formation[49,50]. Phosphorylation of S157 enables VASP translocation to the plasma membrane at the leading edge of migrating cells[51,52] and regulates anti-capping activity[50]. In bFGF-stimulated cells, VASP accumulated in the cortical cytoskeleton and the plasma membrane (Fig. 6d), and in cells invading the brain tissue, we observed filopodia-like protrusions only in control but not in MAP4K4 or STRN3-depleted cells (Fig. S9C). Consistently, depletion of MAP4K4 or of STRN3 caused a robust reduction of bFGF-induced VASP$_{S157}$ phosphorylation in cells seeded on collagen I matrix (Fig. 6e). Under the same condition, VASP$_{S157}$ phosphorylation is repressed by the MAP4K4 inhibitor GNE-495 in DAOY and HD-MBO3 cells (Fig. S9D, E), supporting the notion that MAP4K4 either enables or directly mediates phosphorylation of this residue under basal condition and in response to FGFR activation. Overexpression of full-length MAP4K4 increased bFGF-induced VASP$_{S157}$ phosphorylation (Fig. 6f). No increased VASP$_{S157}$ phosphorylation was observed in cells overexpressing a C-terminally truncated MAP4K4 (KD-ID1-ID2-FLAG, Fig. 1h), or in cells expressing full-length MAP4K4 in the absence of STRN3 (Fig. 6f, g). To test whether the MAP4K4-STRN3 complex can directly phosphorylate recombinant VASP, we performed an in vitro kinase assay using bacterially expressed VASP protein. MAP4K4 immunoprecipitated from bFGF stimulated cells caused increased VASP$_{S157}$ phosphorylation in vitro

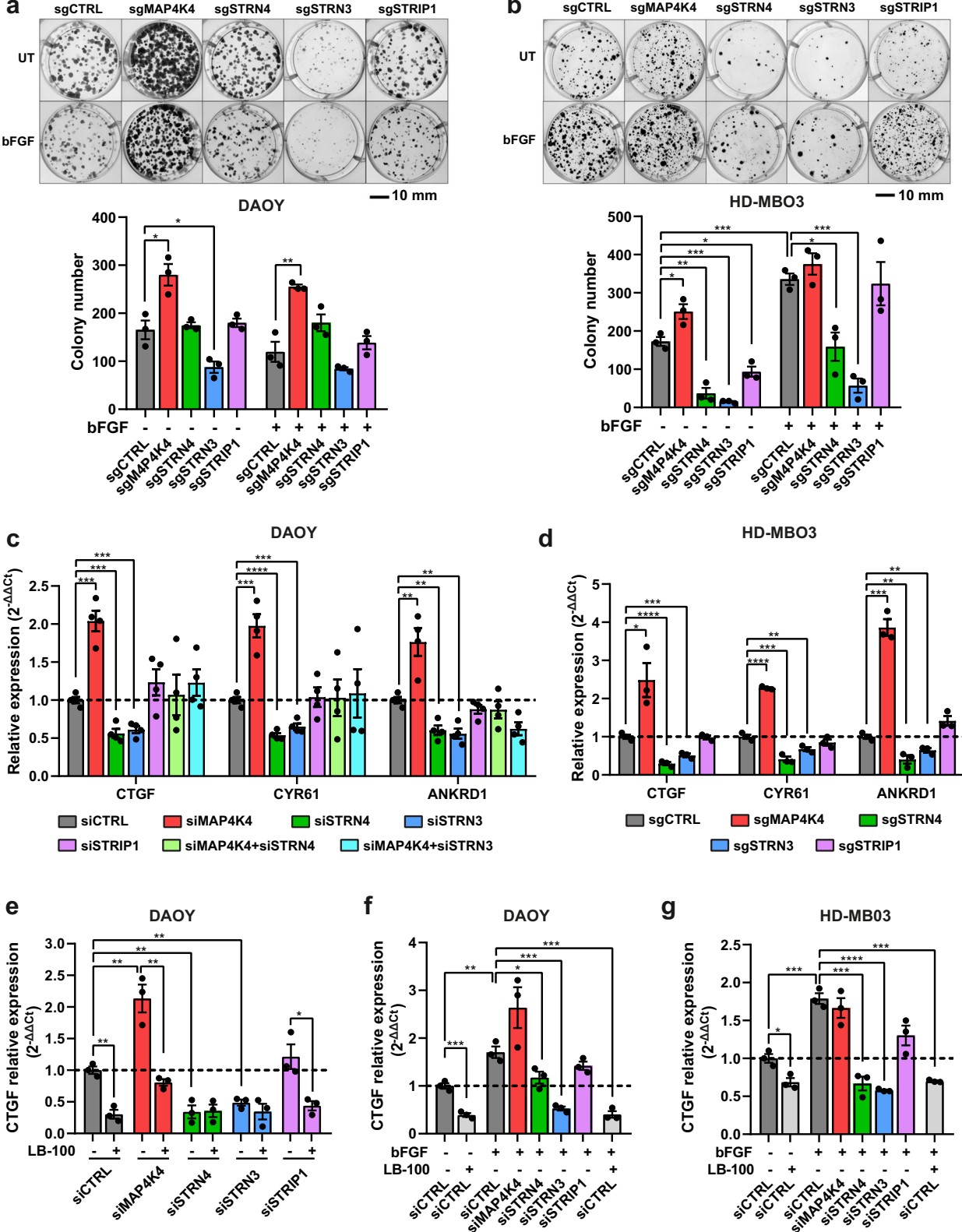

(Fig. 6g). We observed no increase in VASP$_{S157}$ phosphorylation when MAP4K4 is immunoprecipitated from STRN3-depleted cells, suggesting that STRN3 is necessary to either activate MAP4K4 or to couple it to the VASP protein (Fig. 6g). To test the implication of VASP in bFGF-induced invasiveness in MB cells, we assessed siRNA-depletion of VASP using the SIA (Fig. S9F, G). We found

that VASP depletion causes a significant but only partial reduction of bFGF-induced collagen invasion (Fig. 6h).

Taken together, these data identified VASP$_{S157}$ phosphorylation as a novel direct effector mechanism of MAP4K4 and STRN3 downstream of FGFR signaling and highlighted VASP as a mediator of bFGF-induced MB cell invasion.

**Fig. 4 STRN3/4 and MAP4K4 display opposing functions towards MB cell proliferation, clonal growth, and YAP/TAZ target gene expression.**
Representative images and quantification of colony formation assay of DAOY (**a**) and HD-MBO3 cells (**b**) KO for MAP4K4, STRN4, STRN3, or STRIP1 ± 100 ng/ml bFGF ($n = 3$, means ± SEM). **c** qRT-PCR analysis of YAP/TAZ target genes in DAOY cells 48 h after transfection with siRNAs as indicated. ($n = 4$, means ± SEM). **d** qRT-PCR analysis of YAP/TAZ target genes in HD-MBO3 with CRISPR/Cas9-mediated knockout of MAP4K4, STRN4, STRN3, or STRIP1 ($n = 3$, means ± SEM). **e** qRT-PCR analysis of *CTGF* expression in DAOY cells 48 h after transfection with the indicated siRNA ± treatment for 4 h with 5 μM LB-100 ($n = 3$, means ± SEM). qRT-PCR analysis of *CTGF* expression in DAOY (**f**) or HD-MBO3 cells (**g**) transfected with siRNA for 48 h, serum-starved overnight, and treated with 100 ng/ml bFGF for 1 h and/or 5 μM LB-100 for 4 h where indicated ($n = 3$, means ± SEM). *$p < 0.05$, **$p < 0.01$, ***$p < 0.001$, ****$p < 0.0001$ (unpaired Student's *t*-test). See also Fig. S6. Source data for quantifications shown in (**a–g**) are available in Supplementary Data 8.

## Discussion

We found that MAP4K4 interacts with the STRIPAK complex in MB tumor cells, and that this interaction orchestrates growth and invasion control downstream of FGFR signaling (Fig. 7). Mechanistically, MAP4K4 interaction with STRN3 and STRN4 represses canonical MAP4K4 activity towards Hippo tumor suppressor pathway activation by bringing the kinase in proximity of PP2A. Conversely, STRN3 and MAP4K4 interaction is necessary for the activation of PKCθ and VASP, which we found to promote invasion downstream of activated FGFR in MB. This axis of STRIPAK complex function is not repressed by PP2A. We furthermore revealed direct phosphorylation of VASP$_{S157}$ by MAP4K4 and found that this phosphorylation depends on MAP4K4 interaction with STRN3 via the CNH domain. Thus, MAP4K4-mediated activation of PKCθ and VASP constitute a molecular link between the STRIPAK complex and the control of cytoskeleton dynamics. With this, the STRIPAK complex component STRN3 balances the dichotomous functionality of MAP4K4 towards increased invasion and reduced growth repression in MB and other FGFR-driven tumors.

The interaction of the STRIPAK complex with MAP4K4 confirms previous findings describing a direct interaction of STRIPAK with MST1/2, MST3/4, and MAP4Ks[19,25,33,34] in a brain tumor, where MAP4K4 and STRN3/4 are highly expressed. MAP4K4 is a Hippo pathway kinase that through the direct phosphorylation of LATS1/2 caused repression of YAP/TAZ transcriptional activity. Our study confirms the canonical function of MAP4K4 and of STRIPAK towards hippo signaling[21,40,41,53] and links it to increased colony formation and clonal growth in two different MB tumor cell models. From these findings, we conclude that MAP4K4 and STRN3/4 have opposing functions towards Hippo signaling in MB cells, and that the STRIPAK complex exerts a growth-promoting function by suppressing the corresponding activity of MAP4K4 through the phosphatase PP2A. Due to the lack of available cell lines from all subgroups of MB and the inherent limitations of the cell models used to fully represent MB pathogenesis, further studies will be needed generalize our findings to all MB subgroups. MAP4K4 is highly expressed predominately in the SHH subgroup[10], suggesting that STRIPAK repression of MAP4K4-Hippo signaling may have a greater impact on tumor cell functions in this subgroup.

Parallel to antagonistic proliferation regulation of MAP4K4 and STRN3/4, we discovered that MAP4K4 and STRN3 display a co-operative function towards bFGF-induced invasion control. We identified AGC kinases and, in particular, novel PKCs (nPKCs), which are known promoters of tumorigenesis[54], as downstream effectors of MAP4K and STRN3 in invasion control, complementing previous reports of tumor invasion control by STRIPAK with mechanistic insight[26,55]. bFGF-driven invasion in MB may additionally involve the activation of adhesion and cytoskeleton-associated proteins like ezrin-radixin-moesin (ERM), β1 integrin, and focal adhesion kinase (FAK)[56–60], the activity of latter we also found increased by bFGF in a MAP4K4

and STRN3/4-dependent manner. The activation of nPKCs can be induced downstream of RTKs through phospholipase Cγ (PLCγ) signaling[61]. PLCγ is activated by its association with the phosphorylated Y766 residue of the activated FGFR[47]. Depletion of either MAP4K4 or of STRN3 not only decreased the nPKC activity signature but also decreased the phosphorylation of the Y766 residue on FGFR1. This indicates that regulation of Y766 phosphorylation could mechanistically link MAP4K4-STRIPAK function to PKC activation downstream of bFGF stimulation. Specific targeting of PKCδ and PKCθ may thus contribute to an effective blockade of FGFR-driven invasiveness in MB and other FGFR-driven tumors.

STRN3 and MAP4K4 are necessary for maintaining an elongated, mesenchymal motility mode in the brain tissue, and cells devoid of STRN3 or of MAP4K4 display a rounded morphology that expresses fewer filopodia-like protrusions. The involvement of MAP4K4 and STRIPAK complex members in regulating the actomyosin cytoskeleton has been addressed in several studies[11,24]. However, the interplay of MAP4K4 and the STRIPAK complex in cytoskeleton remodeling towards cancer cell dissemination has not been addressed and a direct substrate of MAP4K4 in this process has not yet been identified. Therefore, we consider the phosphorylation of VASP$_{S157}$ by MAP4K4 particularly relevant, because phosphorylation of this residue could couple STRIPAK complex function to VASP localization, cytoskeleton regulation, and cell migration[50,51]. MAP4K4-dependent VASP$_{Ser157}$ phosphorylation may contribute to GF-induced MB cell motility and invasion by promoting cytoskeleton remodeling and the formation of membrane protrusions. In support of this is our observation in previous studies[6] and herein that tissue-invading MB tumor cells display filopodia-like protrusions, and that loss of these protrusions is associated with reduced invasion. MAP4K4 phosphorylation of VASP$_{Ser157}$ in vitro depends on MAP4K4 co-purification with STRN3. Additionally, increased VASP$_{Ser157}$ phosphorylation by MAP4K4 in cells requires the CNH domain of MAP4k4, which enables MAP4K4-STRIPAK interaction. Together, these observations strongly argue for VASP being a direct substrate of MAP4K4 and STRN3 being necessary for productive VASP phosphorylation by MAP4K4. VASP was pulled down specifically by N-BioID2-M4K4 but not C-BioID2-M4K4. This indicates that VASP localizes in the proximity of the kinase domain of MAP4K4 in cells without directly interacting with MAP4K4 and that STRN3 may bring VASP and MAP4K4 in proximity. MAP4K4 and STRN3 also promote the activity of several AGC kinases that are implicated in VASP$_{Ser157}$ phosphorylation, like PKG and PKD1[62], which could act as additional VASP kinases downstream of MAP4K4 in MB cells. Depletion of VASP caused a moderate reduction (−25%) of bFGF-induced collagen I invasion. This indicates that VASP is unlikely the main driver of invasion control in MB. However, additional functions of VASP in tumorigenesis argue for further clarification of the implication of VASP$_{S157}$ phosphorylation by MAP4K4 in MB.

Depletion of STRIP1 moderately increased GF-induced migration and tissue invasion in MB. The increase in the

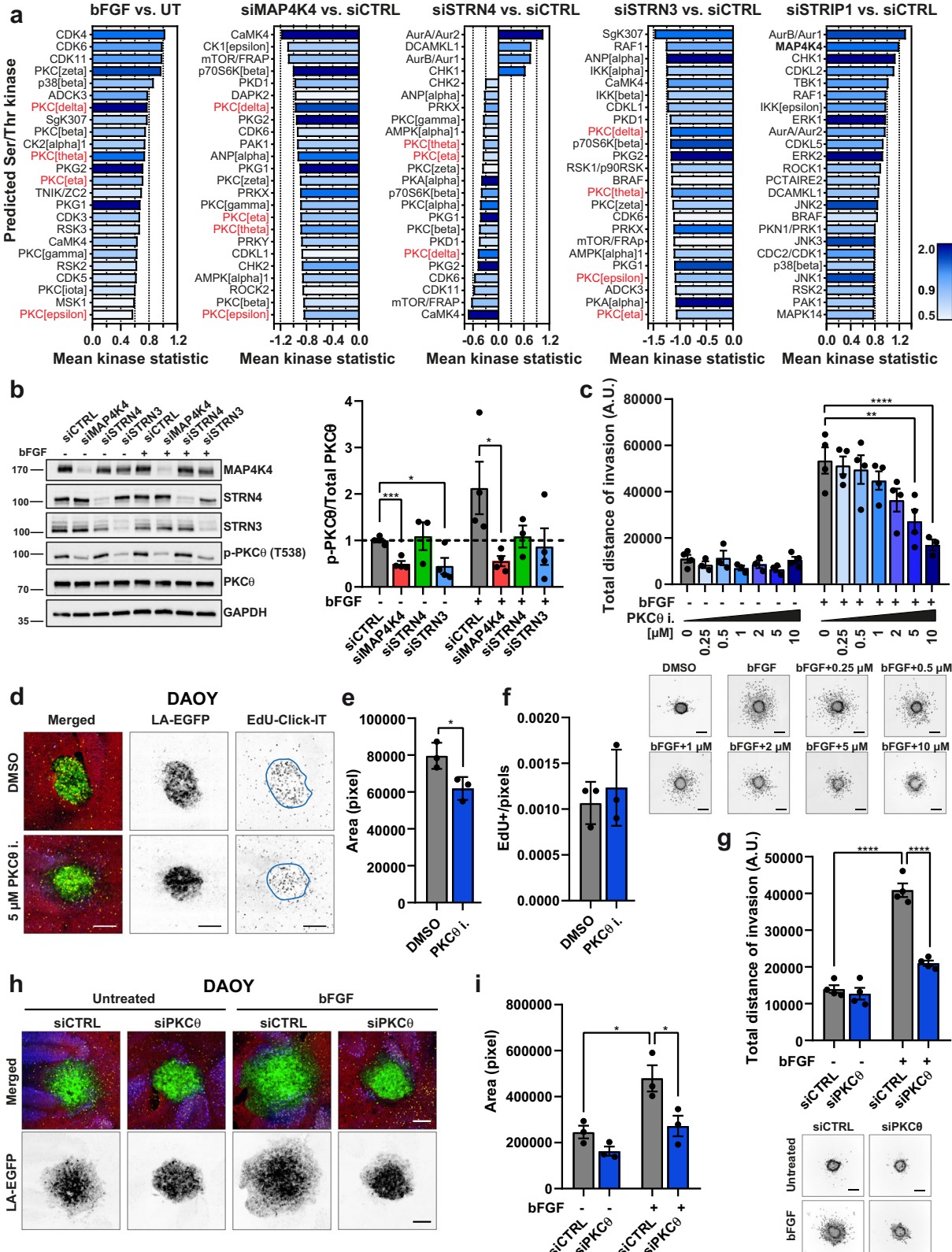

MAP4K4 kinase activity signature after STRIP1 depletion further supports the notion of an antagonizing function of STRIP1 towards MAP4K4. However, the lack of a clear phenotype after STRIP1 depletion also indicates only a minor contribution to migration and growth control in MB tumor cells. STRIP1 depletion has been associated with a contractile motility mode, supporting migration in confined, stiff spaces[63]. Rounded,

blebbing motility may contribute to brain invasion, albeit at lower efficacy, and we previously observed a similar phenomenon in cells treated with the SRC-BCR/ABL inhibitor dasatinib, which represses mesenchymal migration[64].

Together, our data revealed that the cooperation of MAP4K4 and STRN3 promotes motility in a 3D environment and in the brain tissue. This motility depends on the activation of novel

**Fig. 5 Novel protein kinases C are pro-migratory effectors of MAP4K4 and STRN3. a** Kinase activity prediction analysis in DAOY cells transfected with the indicated siRNAs and ± stimulation with 100 ng/ml bFGF for 15 min. The plots show the predicted top 25 differentially activated serine/threonine kinases (STK). The *x*-axis indicates differences in the activity of the predicted protein kinase between the two groups, with effect size (values) and direction (>0=activation; <0=inhibition). The color of the bars represents the specificity score (darker to light is higher to lower score). Values >0.9 were considered as statistically relevant. First graph from left: comparison of bFGF stimulated siCTRL cells vs. untreated (UT); all other graphs: comparison of the indicated siTarget vs. siCTRL with bFGF stimulation. Highlighted in red are novel PKC isoforms (δ, ε, η, and θ) (*n* = 3). **b** IB analysis and quantification of PKCθ $T_{538}$ phosphorylation in DAOY cells 72 h after siRNA transfection ± 100 ng/ml bFGF for 15 min (*n* = 4, means ± SEM). **c** Quantification and representative images of SIA with DAOY cells stimulated with 100 ng/ml bFGF and treated with increasing concentrations of PKCθ inhibitor. Invasion was quantified as the sum of invasion distances from the center of the spheroids. DMSO was used as negative control (*n* = 4, means ± SEM). Scale bar: 300 μm. **d** Maximum intensity projections (MIP) of representative confocal sections of OCSCs implanted with DAOY tumor cell spheroids and treated with 5 μM PKCθ inhibitor or DMSO for five days. Green: Lifeact-EGFP; blue: calbindin (Purkinje cells); red: GFAP; yellow: Edu-Click-IT. Scale bar: 200 μm. **e** Quantification of LA-EGFP area of DAOY spheroids shown in (**d**) (*n* = 3, means ± SD). **f** Quantification of number of EdU-positive nuclei normalized by the area of DAOY spheroids shown in (**d**) (*n* = 3, means ± SD). **g** Quantification and representative images of SIA of siCTRL or siPKCθ transfected DAOY cells ± 100 ng/ml bFGF (*n* = 4, means ± SEM). Scale bar: 300 μm. **h** MIP of representative confocal sections of OCSC implanted with transfected DAOY tumor cell spheroids and treated with 12.5 ng/ml bFGF for 48 h. Green: Lifeact-EGFP; blue: calbindin (Purkinje cells); red: GFAP; yellow: Edu-Click-IT. Scale bar: 200 μm. **i** Quantification of LA-EGFP area of DAOY spheroids shown in (**h**) (*n* = 3, means ± SD). Statistical analyses in **c** and **g** were performed by one-way ANOVA, in **b**, **e**, **f**, and **i** by unpaired Student's t-test. $*p < 0.05$, $**p < 0.01$, $***p < 0.001$, $****p < 0.0001$. See also Figs. S7 and S8. Source data for quantifications shown in (**a–c**), (**e–g**), and (**i**) are available in Supplementary Data 9.

PKCs and on the phosphorylation of VASP, which in concert with the previously identified regulation of ERM proteins and the Arp2/3 complex by MAP4K4[10,17,18], may promote invasive motility in environments of variable complexity and stiffness.

In conclusion, we found that MAP4K4 exerts pro-invasive and anti-proliferative activities in MB cells and that STRN3 contributes to and modulates these MAP4K4 functions downstream of FGFR signaling. This dichotomous functionality of MAP4K4 in tumor cells provides an explanation why MAP4K4 depletion alone is not sufficient to shrink the tumor volume ex vivo, despite the anti-invasive activity of this treatment. Remarkably, however, co-depletion of MAP4K4 and STRN3 caused a near-complete eradication of the tumor cells and completely abrogated cell dissemination in vitro. These findings argue strongly for the development of STRN3 targeting strategies as therapeutic interventions to block MB cell proliferation and invasion. Peptide disruption of the STRN3-PP2A interaction has anti-tumor effects by impairing the phosphatase inactivation of the kinase in the complex and concomitant hippo pathway activation[65]. However, PP2A inhibition repressed invasion in MB cells in a dose-dependent manner, indicating that STRIPAK-associated PP2A is rather unlikely to repress MAP4K4 pro-invasive activity. Rather, targeting of the association of STRN3 and MAP4K4 by impairing CNH-STRN3 interaction could repress the pro-invasive activities of MAP4K4 and unleash its anti-proliferative functions through Hippo pathway activation and thereby impair both MB cell proliferation and dissemination. YAP/TAZ target gene expression and the phosphorylation status of their common downstream effectors PKCθ and VASP in patient samples may inform clinicians in the future about the potential susceptibility to therapies targeting this pathway.

## Material and methods

**Reagents.** Protein phosphatase PP2A inhibitor LB-100 (S7537), PKC-θ inhibitor (S6577), Darovasertib (S6723), Rottlerin (S7862), and Bisindolylmaleimide I (S7208) were purchased from Selleckchem, Houston, TX, USA. MAP4K4 inhibitor GNE-495 was kindly provided by Genentech Inc.[66]. EGF (100-47) and bFGF (100-18B) were purchased from PeproTech, London, UK. If not otherwise indicated, the cells were treated with growth factors at these concentrations: EGF: 30 ng/ml, bFGF: 100 ng/ml.

**Cells and cell culture.** DAOY human MB cells were purchased from the American Type Culture Collection (ATCC, Rockville,

MD, USA) and cultured in Iscove's Modified Dulbecco's Medium (IMEM, Sigma). UW228 cells were generously provided by John Silber (Seattle, USA) and cultured in Dulbecco's Modified Eagle Medium (DMEM, Sigma). HD-MBO3 group 3 MB cells were generously provided by Till Milde (DKFZ, Germany) and cultured in RPMI medium (Sigma). Human embryonic kidney HEK-293T cells were cultured in Dulbecco's modified Eagle's medium (DMEM, Sigma). Cell culture media were supplemented with 10% fetal bovine serum (FBS, Sigma), 1% Penicillin-Streptomycin (ThermoFisher Scientific), and 1% GlutaMAX (Gibco). DAOY LA-EGFP and HD-MBO3 LA-EGFP cells were produced by lentiviral transduction of DAOY and HD-MBO3 cells with pLenti-LA-EGFP. The cells were maintained at 37 °C in a humidified atmosphere containing 5% $CO_2$. All cells were confirmed to be mycoplasma negative. Cell line authentication and cross-contamination testing were performed by Multiplexion by single nucleotide polymorphism (SNP) profiling.

**Plasmids.** Lentiviral constructs were ordered from VectorBuilder (Santa Clara, CA, USA). For BioID experiments, 3xFLAG-tagged BioID2 biotin ligase was fused to either the N-terminus (3xFLAG-BioID2-MAP4K4) or the C-terminus (MAP4K4-BioID2-3xFLAG) of human MAP4K4 cDNA (NM_145686.4). An extended flexible linker consisting of 13 repeats of GGGGS was inserted between MAP4K4 and the BioID2 ligase[27]. As negative controls, only BioID2-FLAG was used. Details of the vectors can be found at https://en.vectorbuilder.com/vector/VB180410-1463qqf.html (N-BioID-MAP4K4), https://en.vectorbuilder.com/vector/VB180410-1356dmp.html (C-BioID2-MAP4K4), and https://en.vectorbuilder.com/vector/VB180411-1121jky.html (BioID2-CTRL). See also Figure S1. For the analysis of MAP4K4 domains, we designed different constructs encoding 3xFLAG-tagged truncated versions of MAP4K4 cDNA (Fig. 1h). Details of the vector can be found at: https://en.vectorbuilder.com/vector/VB190724-1021fsg.html (full-length MAP4K4), https://en.vectorbuilder.com/vector/VB190724-1072aza.html (ID1-ID2-CNH, residues 290-1273), https://en.vectorbuilder.com/vector/VB190724-1048drb.html (KD, residues 1–289), https://en.vectorbuilder.com/vector/VB190724-1089brg.html (KD-ID1-ID2, residues 1–954), https://en.vectorbuilder.com/vector/VB190724-1058sct.html (ID1, residues 290–634), and https://en.vectorbuilder.com/vector/VB190724-1062xrk.html (ID2, residues 635–954). All plasmids were verified by Sanger sequencing.

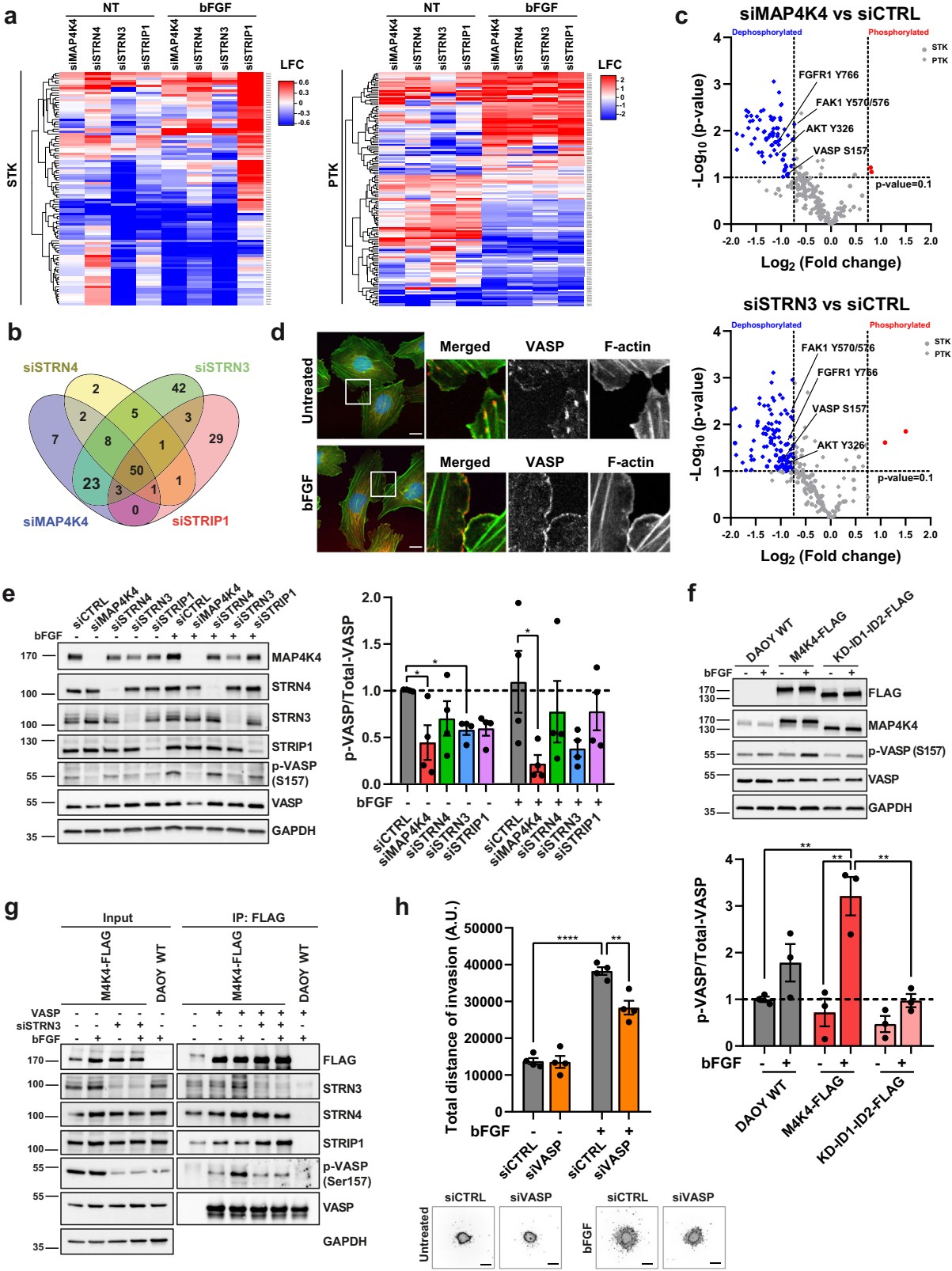

**Mouse maintenance.** Mouse protocols for organotypic brain slice culture were approved by the Veterinary Office of the Canton Zürich (Approvals ZH134/17, ZH116/20). Wild type C57BL/6JRj pregnant females were purchased from Janvier Labs and were kept in the animal facilities of the University of Zürich Laboratory Animal Center.

**Immunoblotting (IB).** To analyze total cell lysates by immunoblot, $1 \times 10^5$ DAOY wild type or $2.5 \times 10^5$ HD-MBO3 cells were lysed using RIPA buffer supplemented with protease (Complete Mini, Roche) and phosphatase inhibitors (PhosSTOP, Roche) and cleared by centrifugation. Protein concentration was assessed using the Pierce BCA Protein Assay Kit (Thermo Fisher

**Fig. 6 MAP4K4 and STRN3 mediate VASP$_{S157}$ phosphorylation. a** Heat map of increased (red) and decreased (blue) phosphorylation of Ser/Thr (STK, left) or Tyr (PTK, right) consensus peptides associated with lysates from siRNA transfected DAOY cells treated ± 100 ng/ml bFGF for 15 min. $Log_2$ of the fold change between siTarget and siCTRL ($n = 3$). **b** Venn diagram displaying the number of Ser/Thr and Tyr peptides with significantly changed phosphorylation ($p < 0.1$) in each siTarget versus siCTRL with bFGF stimulation ($n = 3$). **c** Volcano plots representing the changes in phosphorylation of STK (circles) and PTK (diamonds) peptides in siMAP4K4 (upper) or siSTRN3 (lower) versus siCTRL with bFGF stimulation. Blue: peptides with significantly decreased (fold change < 0.6), red: peptides with significantly increased phosphorylation (fold change > 1.67). The $p$-values were calculated versus siCTRL by ANOVA and post hoc Dunnett's test in the BioNavigator software. $n = 3$. **d** Single confocal section of VASP localization in DAOY cells seeded on collagen-coated plates ± 100 ng/ml bFGF for 15 min. 3× magnifications of boxed areas are shown. Red: VASP, Green: Lifeact-EGFP. Blue: DNA Hoechst. Scale bar: 20 μm. **e** IB and quantification of VASP$_{S157}$ phosphorylation in DAOY cells 72 h after siRNA transfection ± 100 ng/ml bFGF for 15 min ($n = 4$, means ± SEM). **f** IB and quantification of VASP$_{S157}$ phosphorylation in DAOY cells expressing full-length or truncated (KD-ID1-ID2) MAP4K4-FLAG ± 100 ng/ml bFGF for 15 min ($n = 3$, means ± SEM). **g** In vitro VASP S157 phosphorylation of FLAG-immunoprecipitated DAOY cells expressing FLAG-tagged MAP4K4 and transfected for 48 h with the indicated siRNA ± 100 ng/ml bFGF for 15 min. The in vitro kinase reaction was conducted incubating the IP fraction with a recombinant VASP protein. Parental WT DAOY cells were used as a negative control for the IP. **h** Quantification and representative images of SIA of siCTRL or siVASP transfected DAOY cells ± 100 ng/ml bFGF ($n = 4$, means ± SEM). Scale bar: 300 μm. Statistical analysis in **e**, **f** was performed by unpaired Student's t-test, in **h** by one-way ANOVA. *$p < 0.05$, **$p < 0.01$, ****$p < 0.0001$. See also Fig. S9. Source data for quantifications shown in (**e**), (**f**), and (**h**) are available in Supplementary Data 10.

Scientific) according to the manufacturer's instructions. Protein separation was performed on Mini-Protean TGX (4–15% or 4–20%) SDS-PAGE gel (Bio-Rad) and transferred to PVDF membranes (Bio-Rad). After 1 h of blocking with 5% non-fat milk, membranes were probed with primary antibodies listed in supplementary table 1. GAPDH or β-tubulin were used as internal loading control. HRP-linked secondary antibodies (1:5000) were used to detect the primary antibodies. Chemiluminescence detection was carried out using ChemiDoc Touch Gel and Western Blot imaging system (Bio-Rad). The integrated density of immuno-reactive bands was quantified using ImageJ (National Institutes of Health, USA).

To assess PKCθ T538 and VASP S157 phosphorylation, the cells were seeded on a collagen-coated plate and starved for 16 h before treatment with growth factors and inhibitors. Protein extraction was performed by directly incubating the cells at 95 °C for 5 min with a lysis buffer containing 2.5% SDS and 125 mM Tris-HCl pH 6.8 supplemented with protease and phosphatase inhibitors. The lysates were resolved by SDS-PAGE and analyzed by IB as described above.

**Biotin-streptavidin affinity purification.** The protocol for isolating proteins biotinylated by biotin ligase was adapted from the BioID method[27,29]. For proximity-labeling experiments with HEK-293T, $7 \times 10^6$ cells were transiently transfected with N-BioID2-M4K4, C-BioID2-M4K4, or BioID2-CTRL plasmids using jetPRIME reagent (Polyplus Transfection) according to manufacturers' instructions. Non-transfected WT HEK-293T cells were used as a negative control (to exclude biotin-independent binders). Twenty-four hours after the transfection, the cells were incubated with 50 μM biotin for 16 h. For proximity-labeling experiments with DAOY MB cells, we generated cells stably transfected with N-BioID2-M4K4, C-BioID2-M4K4, or BioID2-CTRL lentivirus. $5 \times 10^6$ cells were incubated with biotin as above. Parental WT DAOY cells were used as a negative control. After 16 h of biotin incubation and two PBS washes, the cells were lysed at 4 °C with 1 ml of lysis buffer containing 50 mM Tris-HCl (pH 7.5), 150 nM NaCl, 1 mM EDTA, 1 mM EGTA, 1% Triton X-100, 0.1% SDS and protease inhibitor tablet (Roche). The protein concentrations were normalized among the samples. Two mg of protein samples were incubated with 75 μl of pre-equilibrated Dynabeads MyOne Streptavidin T1 magnetic beads (65602, Thermo Fisher Scientific) in low-binding Eppendorf tubes. The mixtures were incubated for 3 h at 4 °C with end-over-end rotation. The beads were collected using a magnetic rack and washed twice with lysis buffer and twice with 50 mM ammonium bicarbonate (pH 8.3). 10% of the

streptavidin-conjugated beads were used for immunoblotting. Proteins were eluted by heating the beads at 95 °C for 5 min in Laemmli Sample Buffer (1610747, Bio-Rad) with 50 mM DTT followed by magnetic separation. The remaining beads with bound proteins were transferred to a fresh centrifuge tube, resuspend in 150 μl of 50 mM ammonium bicarbonate, and used for MS analysis. For each condition, three biological replicates were performed and analyzed by MS.

**Mass spectrometry (MS).** Tryptic digestion of proteins isolated by streptavidin pull-down was performed based on the protocol described in[28]. Data acquisition was performed by the Functional Genomics Center Zurich (FGCZ). In brief, 50 μl of 8 M urea were added to 150 μl of bead-protein suspensions in 50 mM ammonium bicarbonate (pH 8.3). Beads were reduced by adding 4 μl of 100 mM Tris (2-carboxyethyl) phosphine (TCEP), alkylated with 4 μl of 500 mM chloroacetamide (CAA) and incubated at 30 °C for 1 h in the dark using an Eppendorf Thermomixer at 700 rpm. Sample volume was adjusted by adding 200 μl of digest buffer (10 mM Tris-HCl, 2 mM CaCl$_2$, pH 8.2) to dilute the 8 M urea to 1 M before trypsin digestion. The mixture was treated with 1 μg/sample trypsin and incubated for overnight digestion in the dark at 30 °C using an Eppendorf Thermomixer at 1000 rpm. The supernatant was collected in a fresh Eppendorf tube after magnetic separation of the beads. The beads were washed twice with 100 μl 10 mM Tris-HCl pH 8.2/10% acetonitrile. These washes were pooled with the first eluate and acidified with 60 μl 5% trifluoroacetic acid (TFA, final concentration 0.5%). The samples were desalted on Sep-Pack C18 columns (Thermo Scientific), completely dried using speed vac centrifugation, and dissolved in 20 μl 0.1% formic acid (FA). The samples were diluted 10-fold and transferred to autosampler vials for liquid chromatography-tandem mass spectrometry (LC-MS/MS). 2 μl of samples were injected by an Easy-nLC 1000 system (Thermo Scientific) and separated on an EasySpray-column (75 μm × 500 mm) packed with C18 material (PepMap, C18, 100 Å, 2 μm, Thermo Scientific). The column was equilibrated with 100% solvent A (0.1% FA in water). The column was equilibrated with 100% solvent A (0.1% FA in water). Peptides were eluted using the following gradient of solvent B (0.1% FA in acetonitrile): 5–25% B in, 60 min; 25–35% B in 10 min; 35–99% B in 5 min at a flow rate of 0.3 μl/min. All precursor signals were recorded in the Orbitrap using quadrupole transmission in the mass range of 300–1500 m/z. Spectra were recorded with a resolution of 120,000 at 200 m/z, a target value of 5E5, and the maximum cycle time was set to 3 s. Data dependent MS/MS were recorded in the linear ion trap using quadrupole isolation with a window of 1.6 Da and HCD fragmentation with 30% fragmentation

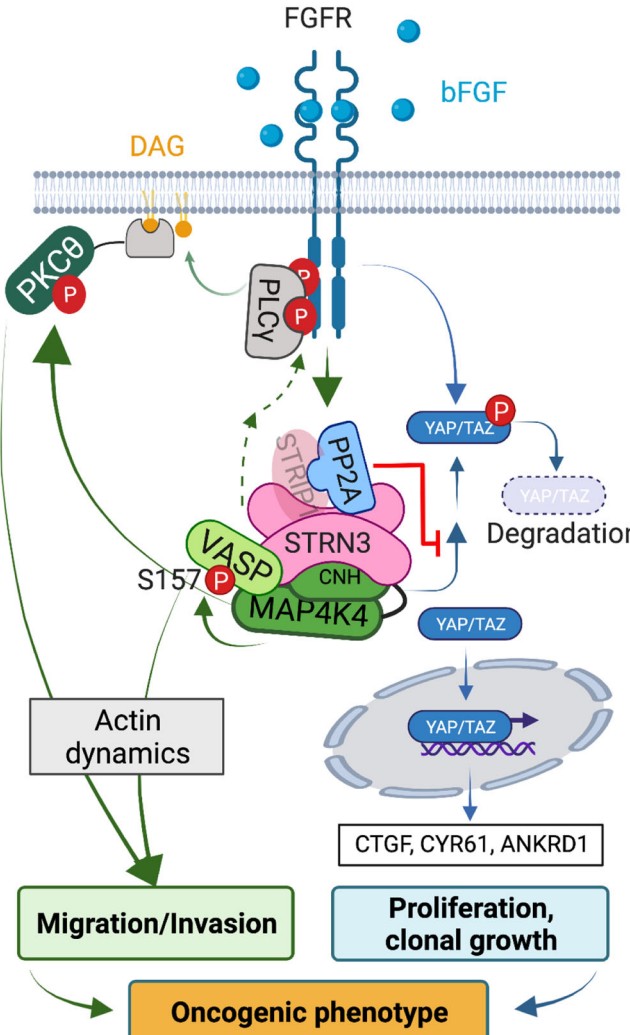

**Fig. 7 Proposed model of MAP4K4 and STRN3 regulation of cell migration and proliferation in MB.** MAP4K4 is activated downstream of RTK signaling and exerts both pro-invasive and anti-proliferative activities in MB cells. Invasion regulation: STRN3 cooperates with MAP4K4 to control cell migration, invasion, and actin cytoskeleton remodeling, independently from PP2A activity. This cooperation of MAP4K4 and STRN3 is necessary for the FGFR-induced activation of motility-associated effectors nPKCs and VASP. Proliferation regulation: MAP4K4 causes LATS1/2 phosphorylation, which promotes the activation of the Hippo pathway. Hippo pathway activation results in the phosphorylation and degradation of the transcriptional regulator YAP. The consequent repression of YAP transcriptional activity leads to a proliferation arrest. STRN3 promotes PP2A-mediated inactivation of MAP4K4 function toward Hippo activation, resulting in an increase of YAP transcriptional activity and cell proliferation.

energy. The ion trap was operated in rapid scan mode with a target value of 8E3 and a maximum injection time of 80 ms. Precursor signals were selected for fragmentation with a charge state from +2 to +7 and a signal intensity of at least 5E3. A dynamic exclusion list was used for 25 seconds. The mass spectrometry proteomics data were handled using the local laboratory information management system (LIMS).

**Proteomic data analysis.** The MS/MS raw data of three biological replicates were combined and searched against a SwissProt database with *Homo sapiens* taxonomy using Mascot (version 2.6.2, London, UK), with a parent ion tolerance of 10 ppm and a fragment ion mass tolerance of 0.030 Da. The database was also expanded with frequently observed contaminants, such as bovine serum albumin, trypsin, and keratins carbamidomethylation of cysteine residues (C) was specified as a fixed modification, and oxidation of methionine (M) was set as a variable modification. Scaffold (version 4.8.9, Proteome Software Inc., Portland, OR) was used to validate MS/MS-based peptide and protein identifications. 1% and 0.1% false discovery rate (FDR) cutoff on peptide and protein, respectively, was used to control for false identifications. Protein identifications were accepted if they contained at least 2 identified peptides per protein. Proteins that contained similar peptides and could not be differentiated based on MS/MS analysis alone were grouped to satisfy the principles of parsimony. Proteins sharing significant peptide evidence were grouped into clusters. Proteins detected in the parental control samples (WT cells lacking BioID2 expression) were subtracted from the results. The significance of enrichment was tested using a two-tailed student's *t*-test with equal variances. Hits were considered significantly enriched in MAP4K4-BioID2 cells compared the unspecific background (BioID2-CTRL) if they had > 2-fold enrichment and p-values < 0.05. Gene ontology (GO) enrichment analysis of MAP4K4-interacting partners was performed using Metascape (http://metascape.org/)[67]. All relevant data have been deposited to the ProteomeXchange Consortium via the PRIDE (http://www.ebi.ac.uk/pride) partner repository with the data set identifier PXD031863 for the MAP4K4 interactome in HEK293T cells and PXD031870 for the MAP4K4 interactome in DAOY cells.

**Co-immunoprecipitation (Co-IP).** $2.5 \times 10^6$ DAOY or $3.5 \times 10^6$ HEK-293T cells were lysed at 4 °C with 400 µl of buffer containing 50 mM Tris-HCl (pH 7.5), 150 nM NaCl, 0.3% NP-40 with protease and phosphatase inhibitor cocktail. Lysates were cleared by centrifugation, and protein concentrations were normalized among the samples. For immunoprecipitation, 30 µl of Dynabeads Protein G magnetic beads (10004D, Thermo Fisher Scientific) were coated with 3 µg of specific antibody or control IgG by resuspending them in 60 µl PBS containing 0.02% Tween-20 (P1379, Sigma), and rotated for 45 min at 4 °C. 1 mg of protein samples was incubated with the antibody-coated Dynabeads for 3 h at 4 °C with end-over-end rotation. The beads were washed four times with TBS buffer (50 mM Tris-HCl pH 7.4, 150 mM NaCl). The proteins were eluted by incubating the beads with 35 µl 100 µg/ml 3xFLAG-Peptide (F4799, Sigma Aldrich) at room temperature for 20 min or by adding Laemmli Sample Buffer (1610747, Bio-Rad) containing 50 mM DTT and denaturation at 95 °C for 5 min. The beads were then magnetically separated from the eluates and the samples analyzed by IB as described above.

**Immunofluorescence analysis (IFA).** To visualize MAP4K4-BioID2 proteins, stably transduced DAOY cells were plated on glass coverslips and incubated with 50 µM biotin for 16 h. The cells were fixed and permeabilized as described in ref. [68]. The fixed cells were incubated with Streptavidin Alexa Fluor 594 conjugate (Invitrogen) for 1.5 h at RT, followed by incubation with primary antibodies overnight at 4 °C. Secondary antibodies were incubated for 2 h at RT.

For VASP localization analysis, 2000 LA-EGFP DAOY cells were seeded in a 384 well plate (781090, Greiner Bio-One) coated overnight with 0.07 µg/µl collagen I (5005-B, Cell systems) diluted in 70% EtOH. After seeding, the cells were starved for 16 h and treated with 100 ng/ml bFGF for 15 min. Cells were fixed and incubated with primary and secondary antibodies as described above.

Imaging acquisition was performed using an SP8 confocal microscope (Leica). The list of the primary and secondary antibodies used is provided in Supplementary Table 1. Metadata of microscopy image acquisition can be found in Supplementary Data 4.

**Lentivirus production and transduction**. Lentiviral particles were produced by transfection of HEK-293T cells with transfer plasmids, psPAX2 (#12260, Addgene) and pCMV-VSV-G plasmids (#8454, Addgene)[69] in a ratio of 5:3:2 using polyethylenimine (24765-2, Polysciences). Virus supernatant was harvested 30 h after transfection, filtered, and concentrated using Amicon Ultra 15 ml Centrifugal Filters (Millipore). For the transduction, human MB cells were incubated for 24 h with supernatant containing the viral particles and 8 μg/ml polybrene. Two days after transduction, cells were selected with puromycin (DAOY: 2 μg/ml, HD-MBO3: 1 μg/ml).

**Generation of CRISPR/Cas9-mediated knockout cells**. Gene-specific single-guide RNA (sgRNAs) were designed using the online design tool at https://design.synthego.com. Only highly specific target sites were selected, and the respective sequences are listed in Supplementary Table 2. Oligonucleotides specific for the target sites were synthesized with BsmBI restriction site overhangs by Microsynth (Balgach, Switzerland) and cloned into the LentiCRISPRv2 transfer plasmid (Addgene plasmid #52961) with a single tube restriction and ligation method as described in ref. [70]. Production of lentiviral vectors and cell transduction was performed as described above. The efficiency of the knockouts was tested by immunoblot.

**RNA interference**. DAOY, UW228, or HD-MBO3 cells at approximately 70% confluency were transfected with Dharma-FECT Transfection Reagent 4 (T-2004-03, Dharmacon) or Lipofectamine RNAiMAX Transfection Reagent (13778075, Thermo Fisher Scientific) with 10 nM siRNAs following the manufacturer's instruction. siRNAs used are listed in Supplementary Table 3. Non-Targeting Control siRNA (siCTRL, D-001210-02-05, Dharmacon) was used as a negative control for siRNA transfection. After 48 or 72 h, RNA and proteins were isolated from the cells to determine gene expression by qRT-PCR and to evaluate protein expression by IB. For SIA or organotypic cerebellum slice culture experiments, the cells were re-seeded to form spheroids in ultra-low adhesion plates 24 h after the transfection.

**RNA extraction and quantitative Real-Time PCR (qRT-PCR)**. Total RNA was isolated from cells with the RNeasy mini kit (74106, Qiagen). cDNA was obtained by retro-transcription of 1 μg total RNA using High-capacity cDNA Reverse Transcription Kit (4368813, Applied Biosystems). qRT-PCR was performed using PowerUp Syber Green (A25776, Thermo Scientific) under conditions optimized for the 7900HT Fast Real-Time PCR System (Applied Biosystems). The primers used were synthesized by Microsynth AG (Balgach, Switzerland). Primer sequences are listed in Supplementary Table 4. The relative expression levels of each gene of interest were calculated using the $2^{-\Delta\Delta Ct}$ method normalized to 18 s.

**Spheroid invasion assay (SIA)**. SIA was performed and quantified according to ref. [32]. In brief: 2500 cells/100 μl per well were seeded in cell-repellent 96 well microplates (650970, Greiner Bio-One, Kremsmünster, Austria). The cells were incubated at 37 °C for 24 h to form spheroids. 70 μl medium was removed and replaced with 70 μl of a solution containing 2.5% bovine collagen

I (CellSystems, Troisdorf, Germany). Polymerized collagen I hydrogels were overlaid with 100 μl of serum-free media containing growth factors and/or inhibitors 2× concentrated. The cells were allowed to invade the collagen matrix for 24 h, after which they were stained with Hoechst (1:2000, B2883, Sigma Aldrich) for 3–4 h. Images were acquired with an AxioObserver 2 mot plus fluorescence microscope (Zeiss, Munich, Germany) or with an Operetta high-content imaging system (PerkinElmer) at 5× magnification. When acquired with the Operetta microscope, the spheroids were localized in the well by acquiring 3 z-planes with 250 μm z-spacing, re-scanned by acquiring 16 z-planes with 25 μm z-spacing and a maximum intensity projection was computed and used for further analysis. Spheroids and invading cells were delineated based on fluorescence threshold. Cell invasion was determined as the sum of the distance invaded by the cells from the center of the spheroid as quantified using an automated cell dissemination counter (aCDc)[32] or using Harmony 4.5 software (PerkinElmer).

**Colony formation assay**. DAOY (400 cells/well) or HD-MBO3 (600 cells/well) cells were seeded in a six-well plate in complete growth medium containing 10% FBS. The cells were allowed to adhere for 48 h, and the medium was replaced with fresh growth medium containing 1% FBS ± bFGF. The cells were cultured at 37 °C for 10 days (DAOY) or 15 days (HD-MBO3) with medium changes every third or fourth day. The colonies were fixed in methanol for 10 min at −20 °C and then stained with 0.5% crystal violet for 15 min. The number and the area of the colonies were quantified by ImageJ using the plugin colony area.

**CellTox green cytotoxicity assay**. Cytotoxicity was determined using the CellTox[TM] Green Cytotoxicity Assay (G8742, Promega, Dübendorf, Switzerland), according to the manufacturer's instructions. 2500 DAOY cells were seeded in 100 μl per well of complete medium in 96-well Corning black wall microplates with clear round bottom (Corning Incorporated, NY, USA). The cells were incubated at 37 °C for 24 h to form spheroids. 50 μl of the medium were then removed from each well and replaced with 50 μL of fresh medium containing 1:500 of CellTox[TM] Green Dye and different drug concentrations. Wells with medium but without cells were used to measure background fluorescence. The fluorescence representing nonviable cells was measured after 24, 48, and 72 h of incubation using Cytation 3 imaging reader (BioTek, Sursee, Switzerland) at Ex/Em 485/520 nm.

**Ex vivo organotypic cerebellum slice culture (OCSC)**. Cerebellar slices were prepared from P8-10 C57BL/6JRj mouse pups[37]. Five to ten wild type C57BL/6JRj male and female mice pups per pregnant female mouse were anesthetized using isoflurane (Forane®, AbbVie Inc., North Chicago, IL, USA, 4–5% isoflurane for 10 min for induction and 2% for 30–40 min for maintenance) at postnatal day 8–10 and then sacrificed by decapitation. Cerebella were dissected and kept in ice-cold Geys balanced salt solution containing kynurenic acid (GBSSK) and then embedded in 2% low melting point agarose gel. Solidified agarose blocks were glued onto the vibratome (VT 1200 S, Leica, Wetzlar, Germany), and 350 μm thick sections were cut (step size: 1 μm; speed: 0.20 mm/s). Slices were transferred on inserts (PICM 03050, Merck Millipore, Burlington, VT, USA) for further in vitro culture and were kept in culture for 15 days. Medium was changed daily, and cultures were monitored for signs of cell death. Tumor spheroids were formed from 2500 DAOY LA-EGFP or HD-MBO3 LA-EGFP cells seeded in ultra-low adhesion plate for 48 h before implantation on OCSCs. Just before implantation, spheroid size and integrity were evaluated under the microscope to

confirm viability, integrity, and similar spheroid size among the different conditions. The slice-spheroid co-cultures were treated with EGF (30 ng/ml), bFGF (12.5 ng/ml), PKCθ inhibitor (5 μM) or Rottlerin (5 μM) for 5 days where indicated. For experiments with siRNA transfected cells, the cells were re-seeded to form spheroids in ultra-low adhesion plate, implanted on the slices 48 h after transfection, and then grown on the slices for other 48 h with or without GF treatment. Following the treatment, the co-cultures were fixed with 4% paraformaldehyde and stained for calbindin, GFAP, or anti-human nuclei, and analyzed using IFA as described in ref. [37]. To evaluate the extent of proliferation both in the tumor cells and in the slice culture, we performed 5-ethynyl-2′-deoxyuridine (EdU) staining using the Click-iT EdU Alexa Fluor 647 imaging kit (C10340, Invitrogen). Four-color image acquisition of three replicates per condition was performed on an SP8 confocal microscope (Leica). The area of the infiltrating spheroids and the number of EdU-positive nuclei were measured with ImageJ (Fiji)[71] using the tool analyzed particles after adjusting the threshold to the same value for all the compared conditions.

**In vitro VASP kinase assay**. $2 \times 10^6$ DAOY cells expressing FLAG-tagged MAP4K4 were transfected with siCTRL or siSTRN3 as described above. 48 h after the transfection and O/N starvation, cells were stimulated with 100 ng/ml bFGF for 15 min and lysed at 4 °C with 400 μl of buffer containing 50 mM Tris-HCl (pH 7.5), 150 nM NaCl, 0.3% NP-40 supplemented with protease and phosphatase inhibitors. Parental WT DAOY cells were used as a negative control. Lysates were cleared by centrifugation, and protein concentrations were normalized among the samples. The supernatants were immune precipitated with 30 μl of Dynabeads Protein G magnetic beads (10004D, Thermo Fisher Scientific) coated with anti-FLAG monoclonal antibody (F1804, Sigma) for 3 h at 4 °C. The beads were washed twice in lysis buffer and twice with kinase assay buffer (20 mM Tris [pH 7.4], 200 mM NaCl, 0.5 mM dithiothreitol, 10 mM MgCl2). The washed beads were resuspended in 25 μl of kinase assay buffer with 100 μM ATP (A7699, Sigma) supplemented with phosphatase inhibitors and incubated for 20 min in an Eppendorf Thermomixer at 30 °C at maximal mixing velocity. Then 4 μg of recombinant human VASP protein (ab105601, Abcam) was added, and the reactions were allowed to continue for 10 min at 30 °C. The kinase reactions were stopped by adding Laemmli Sample Buffer (1610747, Bio-Rad) containing 50 mM DTT and boiling the samples at 95 °C for 5 min. The beads were then magnetically separated from the supernatant, and the samples were analyzed by IB as described above.

**Patient gene expression and proteomic data**. Gene expression and proteomic data of pediatric brain tumor patients were obtained from the ProTrack data portal (http://pbt.cptac-data-view.org/) from the Clinical Proteomic Tumor Analysis Consortium (CPTAC) and the Children's Brain Tumor Tissue Consortium (CBTTC). The dataset used includes a cohort of 218 tumor samples representing seven distinct histological diagnoses, including medulloblastoma (MB), low-grade glioma (LGG), high-grade glioma (HGG), ependymoma (EP), craniopharyngioma (CP), ganglioglioma (GG), and atypical teratoid rhabdoid tumor (ATRT)[31].

**Protein kinase activity profiling**. $4.5 \times 10^5$ DAOY cells were transfected with siRNA as described above. 72 h after the transfection and O/N starvation, cells were stimulated with 100 ng/ml bFGF for 15 min and lysed in M-PER protein extraction reagent (Thermo Fischer Scientific) supplemented with Halt Phosphatase

Inhibitor Cocktail and Halt Protease Inhibitor Cocktail EDTA free (Thermo Fischer Scientific). Lysates were cleared by centrifugation, and protein concentrations were determined by Pierce BCA Protein Assay Kit (Thermo Fisher Scientific). Aliquots of 1 mg/ml of lysates were prepared and snap-frozen in liquid nitrogen. Kinase activity profiles were performed using the PamChip® serine/threonine (STK) and protein tyrosine (PTK) peptide microarray system from PamGene ('s-Hertogenbosch, The Netherlands), according to standard manufacturer's instructions. The PamChip® consists of 4 identical peptide arrays, each array containing 144 (STK) or 196 (PTK) peptides. The technology measures the phosphorylation of the arrayed STK and PTK peptides and compares derived peptide phosphorylation signatures to known consensus signatures of kinases to predict kinase activities. Three PamChip® microarrays were processed at the same time on the PamStation®12 (PamGene). One and five μg of protein extract were used for the STK array and the PTK array protocol, respectively, and the enzymatic assay was started by adding ATP (final concentration 100 μM). For the PTK array, fluorescein isothiocyanate (FITC)–labeled pY20 antibody was incubated on the chip, and the phosphorylation of the individual Tyr peptides was followed by fluorescence detection in real-time. For the STK array, the primary antibody mixture was incubated with the chip, and the secondary FITC- conjugated antibody was used to quantify the signal. For each condition, three biological replicates were performed and analyzed in three different runs.

**Software and data analysis**. The fluorescent signal intensity for each peptide on the PamChip® was quantitated by BioNavigator software (PamGene) and represented as linear regression slope as a function of exposure time. A local background correction was used per spot. Nominal CV (Coefficient of variation) was calculated per peptide with a 2-component error fit model, using the overall mean as input. Only peptides that showed nominal CV < 0.5 were included in the analysis. Additionally, Combat correction/normalization was performed to correct for differences in batch effects. After averaging signal intensities across the three biological replicates per condition, the signal ratio between siTarget and siCTRL was used to calculate fold change (FC) values. Peptides with a FC > 1.67 or FC < 0.6 were considered differentially phosphorylated. To generate the peptide phosphorylation heatmap and compare the global phosphorylation levels, the linear regression slope of each peptide was $\log_2$ transformed. Peptides differently phosphorylated in siTarget vs. siCTRL were identified by ANOVA and postHoc Dunnett's tests. Venn diagram analyses were performed in Venny 2.1 (https://bioinfogp.cnb.csic.es/tools/venny/). An upstream kinase prediction tool based on the PamApp on BioNavigator Analysis software was used to generate a putative list of kinases responsible for phosphorylating the phosphosites on the PamChip. The prediction of the differentially activated upstream kinases for each experimental group (siTarget vs. siCTRL, or bFGF vs. UT) is carried out by a comparison of the phosphorylation sites of the peptides on the array with databases containing experimental and literature-based protein modifications such as HPRD, PhosphoELM (http://phospho.elm.eu.org), PhosphoSite Plus (https://www.phosphosite.org), Reactome (http://reactome.org), UNIPROT (https://www.uniprot.org/) and in silico computational predictions database such as PhosphoNet (http://www.phosphonet.ca/). The prediction tool functional rank-orders the top kinases differentially activated between two compared groups. The mean kinase statistic represents the difference in the predicted protein kinase activity between the two compared groups. A positive value indicates an activation, while a negative value an inhibition. The specificity score indicates the specificity of the

normalized kinase statistics with respect to the number of peptides used for predicting the corresponding kinase. The higher the specificity score, the higher is the power of the prediction. Values of the specificity score >0.9 were considered as statistically relevant. To group the kinases into sequence families, phylogenetic trees were created using the web-based kinome tool CORAL (http://phanstiel-lab.med.unc.edu/CORAL/).

**Statistics and reproducibility.** Statistical analyses were performed using GraphPad Prism 8 software (San Diego, California). Statistical significance of differences between groups was determined using Student's $t$-test (unpaired, two-tailed) or one-way ANOVA repeated-measures test with Bonferroni's correction. The experiments were performed in at least three independent biological replicates ($n = 3$) unless when differently indicated, and results are shown as mean ± standard deviation (SD) or standard error of the mean (SEM). The results were considered significant when $p < 0.05$ (*$p \leq 0.05$, **$p \leq 0.01$, ***$p \leq 0.001$, ****$p \leq 0.0001$). Where indicated, asterisks show statistical significances between control and test sample.

**Reporting summary**. Further information on research design is available in the Nature Research Reporting Summary linked to this article.

### Data availability

The mass-spec data have been deposited to the ProteomeXchange Consortium via the PRIDE (http://www.ebi.ac.uk/pride) partner repository with the data set identifier PXD031863 for the MAP4K4 interactome in HEK293T cells and PXD031870 for the MAP4K4 interactome in DAOY cells. Uncropped immunoblots are shown in Supplementary Information. The MAP4K4 interactome data in HEK293T cells is available in Supplementary Data 1. The MAP4K4 interactome data in DAOY cells is available in Supplementary Data 2. Log fold changes and P values for peptide phosphorylation are available in Ssupplementary Data 3. Metadata of microscopy image acquisition is available in Supplementary Data 4. Source data for all quantifications shown in the figures are available in Supplementary Data 5–17.

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

## Acknowledgements

The authors thank Dr. Savithri Rangarajan (PamGene International) for scientific support and analysis of PamChip imaging data and Dr. Jens Sobek (Functional Genomics Center Zurich) for assistance with the experiments performed on PamStation®12. We would like to thank members of the Functional Genomics Center Zurich for mass spectrometry support. Imaging was performed with equipment maintained by the Centre for Microscopy and Image Analysis, University of Zurich. The illustration in Fig. 7 was created with BioRender.com. This study was supported by grants from the Swiss National Science foundation (SNF_31003A_165860/1, SNF_310030_188793) and the Swiss Cancer League (SCL_KLS-3834-02-2016) to MB, the University of Zürich (Candoc) to J.M. and from the Childhood Cancer Foundation to M.A.G.

## Author contributions

J.M. contributed to designing the study, planned and conducted all the experiments, prepared the figures, and wrote the manuscript. B.Z. generated DAOY cells stably expressing different MAP4K4 domains. C.C. provided experimental support for image acquisition and quantification with the Operetta microscope. M.A.G. helped to draft the study. M.B. designed and drafted the study and wrote the manuscript.

## Competing interests

The authors declare no competing interests.
