## [Peer Review File · Communications Biology]

This manuscript has been previously reviewed at another Nature Portfolio journal. This document only contains reviewer comments and rebuttal letters for versions considered at Communications Biology.

Reviewers' comments:

Reviewer #1 (Remarks to the Author):

The revised manuscript has addressed most of our concerns; we would like the authors to include a statement in the discussion regarding the limitations of the medulloblastoma cell models used and contextualize the results within this framework, in lieu of repeating experiments in further cell lines.

Reviewer #2 (Remarks to the Author):

In this paper, the authors analyzed the kinase activity and phosphorylation sites of multiple STKs/PTKs and found that MAP4K4 interacts with STRN3 to phosphorylate PKC θ and VASP, promoting tumor invasion. They then used PKC θ inhibitor and siRNA of VASP to confirm the role of STRN3-MAP4K4-PKC/VASP axis for tumor invasion, indicating a dichotomous function and specific mechanism of STRN3-MAP4K4 in medulloblastoma growth and invasion.

After reading through the manuscript, the reviewers' comments and the authors' rebuttal, I feel the authors adequately addressed most of raised questions. That said, I noticed the reviewers' interests in specific interaction between STRN3/STRIPAK and MAP4K4. In this regard, the structure and function of STRN3/STRIPAK have been studied relatively in detail (Tang et al., *Cancer Cell* 2020; Tang et al., *Cell Discovery* 2019; Zhang et al., *Structure* 2013). Thus, I recommend the authors take these studies into account, and combine predicted structural information (e.g., AlphaFold/DOCK) to strengthen their conclusion.

Reviewers' comments:

Reviewer #1 (Remarks to the Author):

The revised manuscript has addressed most of our concerns; we would like the authors to include a statement in the discussion regarding the limitations of the medulloblastoma cell models used and contextualize the results within this framework, in lieu of repeating experiments in further cell lines.

We thank the reviewer for this proposition, and we have complemented the discussion with the following statement in red below (second paragraph of discussion):

Lines 294-300:

The interaction of the STRIPAK complex with MAP4K4 confirms previous findings describing a direct interaction of STRIPAK with MST1/2, MST3/4, and MAP4Ks^{19,25,34,35} in a brain tumor, where MAP4K4 and STRN3/4 are highly expressed. MAP4K4 is a Hippo pathway kinase that through the direct phosphorylation of LATS1/2 caused repression of YAP/TAZ transcriptional activity. Our study confirms the canonical function of MAP4K4 and of STRIPAK towards hippo signaling^{30,41,42,54} and links it to increased colony formation and clonal growth in two different MB tumor cell models. From these findings, we conclude that MAP4K4 and STRN3/4 have opposing functions towards Hippo signaling in MB cells, and that the STRIPAK complex exerts a growth-promoting function by suppressing the corresponding activity of MAP4K4 through the phosphatase PP2A. Due to the lack of available cell lines from all subgroups of MB and the inherent limitations of the cell models used to fully represent MB pathogenesis, further studies will be needed generalize our findings to all MB subgroups. MAP4K4 is highly expressed predominately in the SHH subgroup¹⁰, suggesting that STRIPAK repression of MAP4K4-Hippo signaling may have a greater impact on tumor cell functions in this subgroup.

Reviewer #2 (Remarks to the Author):

In this paper, the authors analyzed the kinase activity and phosphorylation sites of multiple STKs/PTKs and found that MAP4K4 interacts with STRN3 to phosphorylate PKC θ and VASP, promoting tumor invasion. They then used PKC θ inhibitor and siRNA of VASP to confirm the role of STRN3-MAP4K4-PKC/VASP axis for tumor invasion, indicating a dichotomous function and specific mechanism of STRN3-MAP4K4 in medulloblastoma growth and invasion.

After reading through the manuscript, the reviewers' comments and the authors' rebuttal, I feel the authors adequately addressed most of raised questions. That said, I noticed the reviewers' interests in specific interaction between STRN3/STRIPAK and MAP4K4. In this regard, the structure and function of STRN3/STRIPAK have been studied relatively in detail (Tang et al., Cancer Cell 2020; Tang et al., Cell Discovery 2019; Zhang et al., Structure 2013). Thus, I recommend the authors take these studies into account, and combine predicted structural information (e.g., AlphaFold/DOCK) to strengthen their conclusion.

We are happy to provide a speculative interaction model based on the so far published structural data combined with the predicted structure of MAP4K4 in AlphaFold. We have included the following scheme in Fig. S2E:

Legend: ... **(E)** Schematic prediction of MAP4K4-STRN3/4 interaction via the citron homology (CNH) domain of MAP4K4 and the WD40 domain of the STRNs (according to ^{54, 73}).

We added (lines 117-119):

STRIPAK complex members STRN4, STRN3, and STRIP1 are highly expressed in MB, and the expression of STRNs correlates positively with site-specific phosphorylation of MAP4K4 in ID1 and negatively with phosphorylation in ID2 *in vivo* (³² and Figure S2A-D). **This suggests that the interaction of these proteins contributes to MAP4K4 regulation and MB pathogenesis, possibly by controlling differential phosphorylation of MAP4K4 (Figure S2E).**